



# Integrated Catchment Classification Across China Based on Hydroclimatological and Geomorphological Similarities Using Self-Organizing Maps and Fuzzy C-Means Clustering for Hydrological Modeling

Jiefan Niu[1,2,3], Ke Zhang[1,2,3,4,5], Xi Li[1,2,3], and Hongjun Bao[4,6]

[1] The National Key Laboratory of Water Disaster Prevention, Hohai University, Nanjing, Jiangsu, 210098, China
[2] Yangtze Institute for Conservation and Development, Hohai University, Nanjing, Jiangsu, 210024, China
[3] College of Hydrology and Water Resources, Hohai University, Nanjing, Jiangsu, 210024, China
[4] China Meteorological Administration Hydro-Meteorology Key Laboratory, Hohai University, Nanjing, Jiangsu, 210024,
China
[5] Key Laboratory of Water Big Data Technology of Ministry of Water Resources, Hohai University, Nanjing, Jiangsu, 210024,
China
[6] National Meteorological Center, China Meteorological Administration, Beijing, 100081, China

*Correspondence to*: Ke Zhang (kzhang@hhu.edu.cn), Hongjun Bao (baohongjun@cma.gov.cn)

**Abstract.** Accurately identifying similar catchments is crucial for transferring model parameters and improving hydrological modeling, especially in ungauged regions with varied climates and topographies. This study presents an integrated method for catchment classification by combining Self-Organizing Maps artificial neural network (SOM) and Fuzzy C-Means clustering (FCM), utilizing hydrometeorological and geomorphological data. We evaluated six climate indices and fifteen landscape characteristics for catchments across China, identifying key variables through correlation and principal component analyses.
The optimal classification produced six distinct climate regions and 35 catchment types with unique streamflow patterns. Validation using ten catchments confirmed the effectiveness of the SOM-FCM approach. The study underscores the importance of considering both climate and landscape factors for a comprehensive classification of catchments, offering valuable insights for hydrological model predictions in ungauged areas and enhancing our understanding of hydrological processes at various timescales.

## 1 Introduction

Runoff prediction is essential for sustainable watershed management at various timescales, including flood defense design, water allocation, and environmental impact assessment (Ma et al., 2021; Wang et al., 2021; Zang et al., 2021). Currently, the models used for runoff prediction more or less rely on observed hydrological data to be calibrated to achieve practically acceptable performance (Liu et al., 2020; Yaseen et al., 2019). However, observed streamflow data are not available for a large
number of catchments throughout the world, which poses a challenge to the application of hydrological models in ungauged catchments (Carozza and Boudreault, 2021; Kratzert et al., 2019). The International Association of Hydrological Sciences



(IAHS) launched the Decade on Predictions in Ungauged Basins (PUB) in 2002 to explore prediction methods for ungauged basins using an improved understanding for climatic and landscape controls during hydrological processes. Over the past few years, the PUB community has proposed a series of regionalization methods, including spatial proximity, physical similarity,

and regression for data-sparse regions(Guo et al., 2021; Kittel et al., 2020; Tsegaw et al., 2019), essentially defining homogeneous zones with similar hydrological characteristics. Among these, the physical similarity approach has become a focal point of research for hydrological forecasting in ungauged areas. It posits that the hydrological response characteristics of a basin are closely related to its climate and underlying surface conditions. Most small and medium-sized basins in China are located in hilly areas, where the lack of monitoring data makes model parameter calibration difficult, leaving them highly

susceptible to natural disasters such as floods and droughts (Liu et al., 2020; Zeng et al., 2021). Therefore, proposing a hydrological regionalization method for small and medium-sized basins is crucial for optimizing model parameters and improving forecasting accuracy in ungauged regions.

Regionalizing a particular hydrological characteristic and applying it to ungauged catchments is complicated as the behavior patterns in hydrology are the consequence of both climate and geomorphology (Gao et al., 2019). It is possible to organize

climatological and geomorphological heterogeneity patterns and develop classification frameworks to categorize catchments with similar hydroclimatological characteristics  (Dallaire et al., 2019; Jehn et al., 2020). This framework assumes that catchments with similar climatic and catchment characteristics also have comparable hydrological behaviors. The increasing availability of climate and geological datasets over the past decade has enabled us to obtain dimensional information and generate new insights into catchment classifications (Addor et al., 2017; Coopersmith et al., 2014). Past efforts to organize

catchments have involved the use of climate characteristics (Knoben et al., 2018; Pagliero et al., 2019), catchment physical features (Leibowitz et al., 2016; Loritz et al., 2019; Tarasova et al., 2020), and hydrological signatures (Addor et al., 2018; Singh et al., 2016). The first two feature types offer the advantage of being available for all geographical locations, and they can be applied directly to ungauged catchments.  Different characteristics are inclined to impact diverse hydrological behaviors (Mcmillan, 2020). According to recent research, climate is the most significant factor related to hydrological behavior,

especially aridity, snow, and seasonality  (Jehn et al., 2020). Another study supporting this conclusion focused on analyzing and clustering 35,215 catchments across Europe. They discovered that the flow signatures were primarily affected by climatic characteristics, particularly those corresponding to average and high flows. Furthermore, topography is also a major factor controlling flow variability (Kuentz et al., 2017). Different spatial scales in research have also led to various interpretations of the key hydrological behavior drivers. At large spatial scales, forcing factors, such as precipitation and temperature, are widely

recognized as having a substantial impact on hydrological processes (Berghuijs et al., 2014a). At small spatial scales, catchments are typically in the same climatic region with similar temperatures and water conditions, and the most significant factor is the hillslope structure of the catchment (Loritz et al., 2018). Recent research on catchment classification has simply combined climate and landscape indices in different climate regions without considering the scale effects of these two types of variables (Ghotbi et al., 2020; Yang et al., 2018). Additionally, some countries have been conducted on hydrological

similarity based on small-sample stations or catchments (Liu et al., 2019; Yang et al., 2020b; Zhai et al., 2021) achieving



significant classification results. However, China has yet to integrate climatic and catchment physical characteristics with spatial scales to propose a comprehensive framework for similar basin classification.

With the advancement of computer technology in the 21st century, the widespread use of machine learning in regionalization studies has become an indisputable fact (Yang et al., 2020a). The self-organizing map (SOM) is an efficient machine learning
method for visualizing complex high-dimensional data structures on a 2D surface (Kohonen, 1982). However, the number of output neurons exceeds the expected number of groups in most applications (Kiang, 2001). As a result, previous studies have added clustering steps to SOM results, such as k-means clustering algorithms and hierarchical clustering algorithms, to generate appropriate groups and facilitate quantitative analysis (Boscarello et al., 2016; Nguyen et al., 2015). Combining these algorithms has been used in various fields. Zang et al. (2021) used SOM with k-means to map future drought conditions in
China, and produced relatively accurate classification results. Kim et al. (2020) investigated deep thermal groundwater in South Korea using SOM combined with hierarchical clustering and found that five major clusters accounted for various bedrock groundwater geochemistry groups. These clustering methods are effective when the sample data clearly define the bounds in the feature space. The fuzzy c-means algorithm (FCM) is a soft clustering algorithm that can accommodate the heterogeneity of data by assuming that each sample belongs to all groups with varying degrees of membership (Bezdek et al.,
1984). Lee et al. (2019) used SOM in conjunction with FCM to assess the overall quality of groundwater in Seoul and classified the water samples into three groups. Combining these two algorithms aggregates the advantages of each algorithm. Based on the self-organization and nonlinear mapping capabilities of the SOM algorithm as well as the concept of the fuzzy set from the FCM algorithm, it can explain the highly heterogeneous and nonlinear data associated with fuzzy boundaries. Research shows that the combined algorithms produced good results in water quality identification and climate clustering applications (Knoben
et al., 2018; Lee et al., 2019). However, there have been no attempts to combine these two algorithms for small catchment hydrological regionalization.

The objective of this study was to combine SOM and FCM algorithms to identify hydrologically similar catchments in China. This research contributes to a better understanding of the relationship between the similarities of hydroclimatological and geomorphological attributes, and the similarities of hydrological processes on the basin level across China.

## 2 Methodology and data

### 2.1 Methods

The methodology integrates hydroclimatological and geomorphological data to identify catchment similarities. This approach combines the Self-Organizing Maps (SOM) algorithm, an unsupervised artificial neural network, with Fuzzy C-Means (FCM) clustering, a soft clustering method rooted in fuzzy set theory. The process begins with the selection and preparation of relevant
climate indices and geomorphological characteristics, followed by the unsupervised classification of catchments. Initially, SOM is used to group catchments based on spatial climate patterns, identifying regions of meteorological homogeneity. Subsequently, FCM refines the classification by clustering catchments within these regions, accounting for gradual transitions





in climate and landscape. Finally, the classifications are validated using streamflow data from selected catchments to ensure the model's reliability.

### 2.1.1 Selection of climate indices preparation

The spatial patterns of climate and landscape have been evaluated as causal factors in determining the hydrological response of catchments. Climate not only directly influences runoff generation processes at the event scale but also indirectly affects the hydrological cycle by acting on longer-term soil moisture availability and the co-evolution of landscape and vegetation. Climate patterns produce significant differences in the long-term balance between available water and energy. Catchments in arid regions are generally considered to have a limited water supply. It is characterized by sparse precipitation with high evaporation, resulting in less recharge of the aquifer. In addition, the stream in these regions is usually lost and reaches a certain position, with the flow infiltrating through the riverbed to recharge the underlying aquifer. In contrast, the catchments in humid regions are considered energy-limited, and there is higher precipitation and less episodic infiltration. A portion of groundwater recharges the streamflow through the waterway. It is plausible that climate patterns influence hydrological partitioning in ways reflected in runoff records. An earlier study indicated that five distinct climate indices may be relevant to hydrological processes (Addor et al., 2017; Betterle et al., 2019; Knoben et al., 2018): (1) the annual average aridity, (2) the seasonality of aridity, (3) the fraction of precipitation that falls as snow, (4) the average rainfall intensity, and (5) the seasonality of rainfall intensity. Although the precipitation intensity index values can vary widely throughout the world, their effects on hydrologic processes are mainly determined by local catchment characteristics. Consequently, the precipitation intensity index was not considered in this study, and the influence of landscape differences in watershed characteristics on hydrological processes will be discussed in later sections.

Earlier studies illustrated that the indices of aridity and snow are strongly correlated with streamflow patterns without considering rainfall intensity (Knoben et al., 2018). Temperature, as an indicator of the snow and evapotranspiration process, adds independent information to the discussion of climate similarity. Therefore, three indices were added to capture the seasonal and spatial variability of temperature, based on the original aridity and snow indices. Ultimately, we selected six indices for climate classification: average moisture index ($I_m$), seasonal moisture index ($I_{m,r}$), that falls as snow ($fs$), annual average temperature ($T_m$), seasonal temperature ($T_{m,r}$) and the fraction of snowy days ($Ds$). The first three are expressed using a version of Thornthwaite's moisture index $MI$ (Willmott and Feddema, 1992). These indices were calculated for each $0.25°$ land cell based on the CRU TS V4.04 dataset and the meteorological station data. Some of these indices have been previously used to map climate homogeneity regions, but not in this specific combination. The climate indices were calculated using the following equations:

$$MI(t) = \begin{cases} 1 - \dfrac{E_P(t)}{P(t)}, P(t) > E_P(t) \\ 0, P(t) = E_P(t) \\ \dfrac{P(t)}{E_P(t)} - 1, P(t) < E_P(t) \end{cases}, \tag{1}$$





$$I_m = \frac{1}{12}\sum_{t=1}^{t=12} MI(t), \tag{2}$$

$$I_{m,r} = \max\big(MI(1,2,\dots,12)\big) - \min\big(MI(1,2,\dots,12)\big), \tag{3}$$

$$fs = \frac{\sum P(T(t)\leq T_0)}{\sum_{t=1}^{t=12} P(t)}, \tag{4}$$

$$T_m = \frac{1}{12}\sum_{t=1}^{t=12} T(t), \tag{5}$$

$$T_{m,r} = \max\big(T(1,2,\dots,12)\big) - \min\big(T(1,2,\dots,12)\big), \tag{6}$$

$$Ds = \frac{\sum D(T(t)\leq T_0)}{\sum_{t=1}^{t=12} D(t)}, \tag{7}$$

$P(t)$, $E_P(t)$, and $T(t)$ are the mean monthly observed values of precipitation, potential evapotranspiration, and temperature,
respectively; $D(t)$ is the number of days per month; $T_0$ is the threshold temperature, below which precipitation is presumed to
occur in snow form, set at 0 °C.

### 2.1.2 Self-organizing map clustering algorithm

The function of catchment classification is to map the complex spatial structure of hydrological patterns onto regional patterns
that are immediately recognizable and interpretable. We recast similar hydrological patterns through two-step clustering to
first identify the meteorological homogeneity regions and then classify the catchments within the same climate clustering.
Self-organizing map (SOM) is an artificial neural network model proposed by Kohonen (Kohonen, 1982). The SOM technique
is an unsupervised nonlinear technique capable of mining algorithmic rules embedded in samples. It can automatically identify
the important internal statistical characteristics of samples through competition and interaction among neurons. Owing to its
automatic clustering, nonlinear mapping, and fault tolerance features, this algorithm has been widely used to solve pattern
recognition and classification problems (Jeong et al., 2010; Zhai et al., 2021).
The SOM algorithm, which contains the input and competition layers, projects high-dimensional input data onto low-
dimensional output surfaces composed of an array of ordered neurons. There is complete connectivity between each neuron $i$
and all the input samples, which is represented by p-dimensional weight vectors $\boldsymbol{w}_i = [w_{i1}, w_{i2}, \dots, w_{ip}]^T$ ($p$ is the
dimensionality of the input space). Moreover, adjacent neurons are interconnected through neighborhood relationships. The
SOM neural network performs an iterative procedure called competitive learning, in which neurons are organized according
to their similarities and the best matching unit (BMU) is generated through competition among neurons. The crucial links of
competition, cooperation, and adaptation between neurons occur throughout the entire learning process. We can select the
proper structure of the competitive layer in the SOM algorithm by evaluating the network internal indicators of quantitative
error ($QE$) and topological error ($TE$) as follows (Jeong et al., 2010; Park et al., 2003):

$$QE = \frac{\sum_{i=1}^{N}\|x_i - W_g(x_i)\|}{\sum_{i=1}^{N}\|x_i\|}, \tag{8}$$





$$TE = \frac{1}{N}\sum_{i=1}^{N} u(x_i), \tag{9}$$

where $N$ is the number of samples; $W_g$ is a weight vector of the BMU; $QE$ and $TE$ measure the average relative distance

between the input sample and the corresponding BMU, and the topology retention after sample projection, respectively; $u(x_i)$

is equal to 1 when adjacent samples remain adjacent after projection; otherwise, it is 0. The competitive layer acquires the best

neuron structure as $QE$ and $TE$ converge to the minimum values.

The learning algorithm consists of the following four steps: (1) initializing the structure and parameters of the competitive

layer, (2) calculating and comparing the Euclidean distance between each input sample $x$ (i.e., climate indices and catchment

characteristics, respectively) and every neuron, (3) identifying the closest neuron to the input sample as the BMU of the input,

and (4) updating the BMU and its neighbor's weight vector in response to the input sample. The weight vector at each time

step $t + 1$ as follows:

$$\boldsymbol{w}_i(t+1) = \boldsymbol{w}_i(t) + \partial(t)\delta_{i,j}(t)[\boldsymbol{x} - \boldsymbol{w}_i(t)], i \in \delta_{i,j}(t), \tag{10}$$

where $\partial(t)$ is the learning rate and $\delta_{i,j}(t)$ is the neighborhood kernel function of neuron $i$ for its BMU ($j$), which indicates the

radius that the BMU can influence. By increasing the time step($t$), the $\delta_{i,j}(t)$ value gradually decreases so that the neurons

nearer the BMU are updated to have stronger connections with it. Consequently, neurons adjacent to the BMU are drawn closer

to the BMU than to other neurons, and exhibit greater similarities with the BMU. Following the completion of the iterative

learning, every input sample was assigned to its BMU. Samples with similar properties are grouped in the same BMU, which

is considered the clustering center in the SOM. SOM was calculated and visualized using the SOMPY toolbox in Python.

### 2.1.3 Fuzzy c-means clustering algorithm

Fuzzy c-means clustering (FCM), based on fuzzy set theory, is one of the most widely used soft clustering algorithms. Unlike

hard clustering algorithms, such as k-means and hierarchical clustering, the FCM cluster procedure uses a fuzzy parameter to

create overlapping cluster boundaries. By allocating each sample to all groups with a degree of membership ranging between

0 and 1, undistinctive data can form overlapping clusters with vague boundaries (Pal et al., 2005). As climate and catchment

landscape characteristics show gradual changes rather than distinct changes in space, the FCM algorithm may be the most

suitable method for clustering hydrological regions. Furthermore, it requires a relatively short computation time to generate

180 an appropriate classification with a low sensitivity to initialization. FCM clustering was performed iteratively by minimizing

the following objective functions:

$$F = \sum_{j=1}^{C}\sum_{k=1}^{N} u_{jk}^{m}\left\| x_k - v_j \right\|^2, \tag{11}$$

$$u_{jk} = \frac{1}{\sum_{i=1}^{C}\left(\frac{\left\| x_k - v_j \right\|}{\left\| x_k - v_i \right\|}\right)^{\frac{2}{m-1}}}, \tag{12}$$



where N is the number of samples, $C$ is the number of clusters, $u_{jk}$ is the membership value of sample $k$ in cluster $j$; $x_k$ is the value of sample $k$; $v_j$ is the center value of cluster $j$; and $m$ is the degree of fuzziness in the clustering process.

The number of clusters ($C$) must be defined before performing the partitioning in the FCM algorithm. Various validity indices have been proposed to determine the optimal number of clusters (Halim et al., 2017; Pakhira et al., 2004). Two internal indicators were selected: the Davies-Bouldin index ($DBI$) and silhouette coefficient ($SC$) (Rao and Srinivas, 2006). Both $DBI$ and $SC$ measure the degree of density within clusters and the amount of disorganization between clusters. The optimum number of clusters was selected by minimizing $DBI$ and maximizing $SC$. Each function is represented by the following equations:

$$DBI = \frac{1}{C}\sum_{j=1}^{C} max\left(\frac{\overline{s_i}+\overline{s_j}}{\|v_j - v_i\|_2}\right)$$

$$\overline{S_j} = \left(\frac{1}{n_j}\sum_{x \in C_j}\|x - v_j\|\right)^{1/2},\tag{13}$$

$$SC = \frac{1}{N}\sum_{k=1}^{N}\frac{b(k)-a(k)}{max\{a(k),b(k)\}},\tag{14}$$

where $n_j$ is the number of samples in cluster $j$; $v_j$ is the cluster center matrix of cluster $j$; $\overline{S_j}$ is the average distance between $v_j$ and the others in cluster $j$; $N$ is the number of samples, $a(k)$ is the average distance between sample $k$ and the others in the same cluster, $b(k)$ is the minimum average distance between sample $k$ and each sample in a different cluster.

Based on the output layer of the SOM neural network, the FCM algorithm was used for clustering. The membership values of the neurons belonging to different clusters were assigned to the corresponding samples after grouping every output neuron into clusters. This illustrates the spatial distribution of the climate and landscape characteristics. This method is used for climate partitioning and classification of catchments.

**2.2 Dataset**

Precipitation (P) and temperature (T) datasets were collected from the National Meteorological Information Center (http://data.cma.cn/) at the Chinese Meteorological Administration (CMA). These data were available for 1982-2015 from 613 stations in China. Potential evapotranspiration (EP) data were provided at a 0.5° × 0.5° resolution from the CRU TS V4.04 dataset at the Centre for Environmental Data Analysis (https://www.ceda.ac.uk/). The EP is estimated using a variant of the Penman-Monteith formula. The climate data were interpolated at a 0.25° × 0.25° spatial resolution, and missing data were filled using the weighted nearest-neighbor approach.

The HydroSHEDS dataset from the World Wildlife Fund (https://www.hydrosheds.org/page/overview) comprises a large database of georeferenced information on stream networks, watershed boundaries, and drainage directions (Lehner and Grill, 2013). The HydroBASINS dataset contains seamless coverage of consistently sized sub-basins used to delineate catchments. Preliminary quality assessments have concluded that these data are significantly more accurate than existing data on watersheds and rivers, and have been applied to a variety of studies (Anh and Aires, 2019; Carozza and Boudreault, 2021; Yamazaki et





al., 2014). Therefore, 13,487 catchments from the HydroBASINS dataset in China were analyzed. The catchment was
classified using clustering technology, and hydrologic similarity regions were established based on catchment signatures. The
catchment landscape was characterized by analyzing a wide range of features, which can be organized into three classes:
topographic characteristics, soil and vegetation characteristics, and topological characteristics. The topographic and
topological properties were computed using the digital elevation model ASTER GDEMV2 with a resolution of 30 m, which
was provided by the Geospatial Data Cloud, Computer Network Information Center, Chinese Academy of Sciences
(http://www.gscloud.cn). The soil and vegetation characteristics were derived from the 1:1 million soil map of China at the
Institute of Soil Science (http://www.issas.ac.cn) and the Spot/vegetation NDVI dataset provided by the Resource and
Environmental Science and Data Center (https://www.resdc.cn), respectively.

**Table 1.** Summary of catchment descriptors used in this study.

| | Description | Variables | Unit | Mean | Range |
|---|---|---|---|---|---|
| Topographic characteristics | Mean elevation | $H$ | m | 1,795.52 | -134.36-5,803.84 |
| | Elevation range | $\Delta H$ | m | 1,086.08 | 3-7,315 |
| | Hypsometric curve integral | $HI$ | - | 0.64 | 0.24-1 |
| | Gradient of hypsometric curve | $AS$ | - | 0.52 | 0-1.84 |
| | Mean topographic index | $TI$ | - | 10.02 | 7.43-12.72 |
| | Mean slope | $\beta$ | degree | 5.45 | 0.02-28.32 |
| Soil and vegetation characteristics | Sand fraction | $Sand$ | % | 44.42 | 0-91.72 |
| | Clay fraction | $Clay$ | % | 20.22 | 0-52.23 |
| | Silt fraction | $Silt$ | % | 33.14 | 0-54.00 |
| | NDVI | $NDVI$ | - | 0.56 | 0-0.9 |
| Topological characteristics | Area | $A$ | km$^2$ | 761.04 | 15-14,612.8 |
| | Length | $L$ | km | 52.96 | 9.3-615.7 |
| | Form factor | $Rf$ | - | 0.33 | 0.02-1.97 |
| | Elongation ratio | $Re$ | - | 0.61 | 0.15-1.58 |
| | Drainage density | $Rd$ | km/km$^2$ | 0.29 | 0.06-1.40 |

The complete catchment signatures from the various data sources mentioned above are provided, as well as details on the
statistical characteristics (Table 1). Fifteen characteristics were selected to describe catchment landscapes. These signatures
were calculated for all the catchments with a wide spectrum of characteristics. These data were relatively simple to obtain,
allowing the application of this method to some ungauged basins. They can also improve the prediction of hydrological
signatures (Addor et al., 2018; Addor et al., 2017; Boscarello et al., 2016; Jehn et al., 2020). In addition, we illustrate some of
these characteristics using Eq. (1-3). The integral of the hypsometric curve (HI) indicates the catchment's surface quality, and





the gradient of the hypsometric curve (AS) reflects the degree of topographic relief. Both are calculated using the hypsometric curve f(x). The mean topographic index (TI) is the arithmetic mean of the topographic indices of the raster cells in the catchment.

$$HI = \int_0^1 f(x)dx, \tag{15}$$

$$AS = \frac{f(0.2) - f(0.8)}{0.8 - 0.2}, \tag{16}$$

$$TI = \frac{1}{n}\sum_{i=1}^n \ln\left(\frac{\alpha_i}{\tan\beta_i}\right), \tag{17}$$

where f(0.2) and f(0.8) are expressed as the hypsometric curve's relative elevation differences corresponding to 0.2 and 0.8, respectively; $\alpha$ and $\beta$ indicate the drainage area and slope of the cell in a catchment, respectively; n is the total number of raster cells in the catchment.

## 3 Results

### 3.1 Classification of regions based on climate factors

#### 3.1.1 Spatial distributions of climate indices

Figure 1 shows that the spatial distribution of six individual climate indicators gradually changed. Climate features are primarily influenced by latitude and altitude; however, abrupt changes in topography (e.g., the Tianshan and Himalayan Mountains) can result in relatively sharp climate transitions in some regions. The six indicators were standardized to between 0 and 1, and were then applied to show a single overview of the moisture ($I_m$, $I_{m,r}$, and $fs$) and temperature indices ($T_m$, $T_{m,r}$, and $Ds$) with RGB color scales combined to a map, respectively (Fig. 1g). This visualizes the spatial distribution of each climate indicator and describes them according to their corresponding relationships. Overall, there was a trend from northwest to southeast in the first three indices, indicating the degree of wetness in China. (Fig. 1g, left). Specifically, arid regions (red areas) are located in the northwest part of China and are characterized by a large desert with high PET compared to available precipitation, almost no seasonal changes, and no snowfall. The wet regions (dark green) are concentrated along the middle and lower reaches of the Yangtze River. . Generally, these areas have no snowfall, little seasonal variation, and continuous rainfall. The transitional climate regions (bright green and yellow) are located between arid and humid zones, and the climate in this region experiences strong seasonality in their water-energy balance, most notably in the seasonal variation of precipitation and the seasonal pattern of PET.  The pink regions indicate areas where the majority of the precipitation falls as snow.

Unlike the regional distribution of the wetness indices, the temperature feature gradient from the indices varies gradually along the latitudinal band and mutates with abrupt changes in topography (Fig. 1g, right). There are regions with high temperatures in southern China (dark green), where the temperature remains constant throughout the year.The low-temperature regions



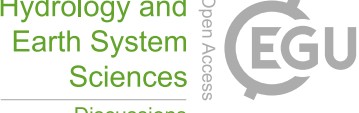

260 (yellow) in northern China have relatively strong seasonal temperature changes and snow processes. The transitional climate regions (dark yellow and green) are located between two zones, increasing in temperature and decreasing in seasonality from north to south. Furthermore, there is a low-temperature region (pink) in the southwest, influenced by the Tibetan Plateau, where snow accumulates for years and temperatures remain extremely low with little seasonal variation.

According to the climate spatial distribution characteristics of each individual indicator, the spatial distribution of 265 meteorological variables has a strong regional component in China. Meanwhile, the spatial distribution of snow days was strongly inversely correlated with the average temperature (Spearman rank correlation coefficient was -0.82). To avoid a classification procedure that could introduce redundancy, the snow day index was removed, and five different climate indices were selected for climate partitioning.

270 **Figure 1.** Maps of average values of climate indices from 1982 to 2015 across China: (a) average moisture index $(I_m)$, (b) seasonal moisture index $(I_{m,r})$, (c) fraction of precipitation that falls as snow $(fs)$, (d) annual average temperature $(T_m)$, (e) seasonal temperature $(T_{m,r})$, and





(f) fraction of snowy days (**Ds**). Panel (g) provides an overview of the climate indices combination using an RGB color scale, where the moisture indices (left) are represented by $I_m$ (red), $I_{m,r}$ (green), and $fs$ (blue), and the temperature indices (right) by $T_m$(red), $T_{m,r}$ (green), and **Ds** (blue).

### 3.1.2 The clustering results of SOM

Using standardized climate index raster data from China, SOM technology was used to cluster climate data, and 418 output neurons were obtained. The neurons are displayed on a 19×22 rectangular grid with 418 hexagons, as shown in Fig. 2. Here, the structure of the neurons was selected based on the network internal indicators of quantization error (QE) and topographic error (TE). Vesanto (1999) suggested that SOM results can be expressed in the form of two types of maps: component planes and distance matrices (d-matrices). On the component planes, the individual neuron weight vector values are shown using color coding; blue and red correspond to low and high values, respectively. This allows the recognition of the mutual dependence relationship among variables when comparing the patterns of the component planes. For instance, opposite gradients in the component planes indicate a negative correlation between the variables. In the median d-matrix, the median Euclidean distances between neighboring neurons are indicated by the color scale. Consequently, we can obtain an indication of the relative distances between neurons; neurons with high similarity (blue) may be considered as clusters.

Five features are apparent in the component planes (Fig. 2a). First, the vector values did not exhibit a horizontal or diagonal gradient distribution in color. Overall, the five climate indicators had different patterns, and the weight vectors in the component planes did not follow a uniform distribution, indicating that the indicators were relatively independent and could represent a range of hydroclimatic characteristics. Second, the low-value regions of $I_m$, $I_{m,r}$, as well as the high-value regions of $I_m$ and $T_m$, have a consistent distribution on the component plane map. The consistency between climate indices indicates that several climate features develop synergistically in local regions. For example, the arid regions exhibit stable humidity with little seasonal variation, while humid regions have high temperatures and precipitation predominantly in the form of rainfall. This aligns with existing studies on the synchrony of rain and heat in China's climate characteristics (Hao et al., 2018).

The data for the climate indices were continuously distributed rather than discretely clustered. It was difficult to distinguish the cluster structures of the data in the d-matrix (Fig. 2b). Therefore, we utilized the FCM algorithm to reveal the underlying cluster structures in SOM neurons, which considers the complexity of climate spatial continuity and heterogeneity.





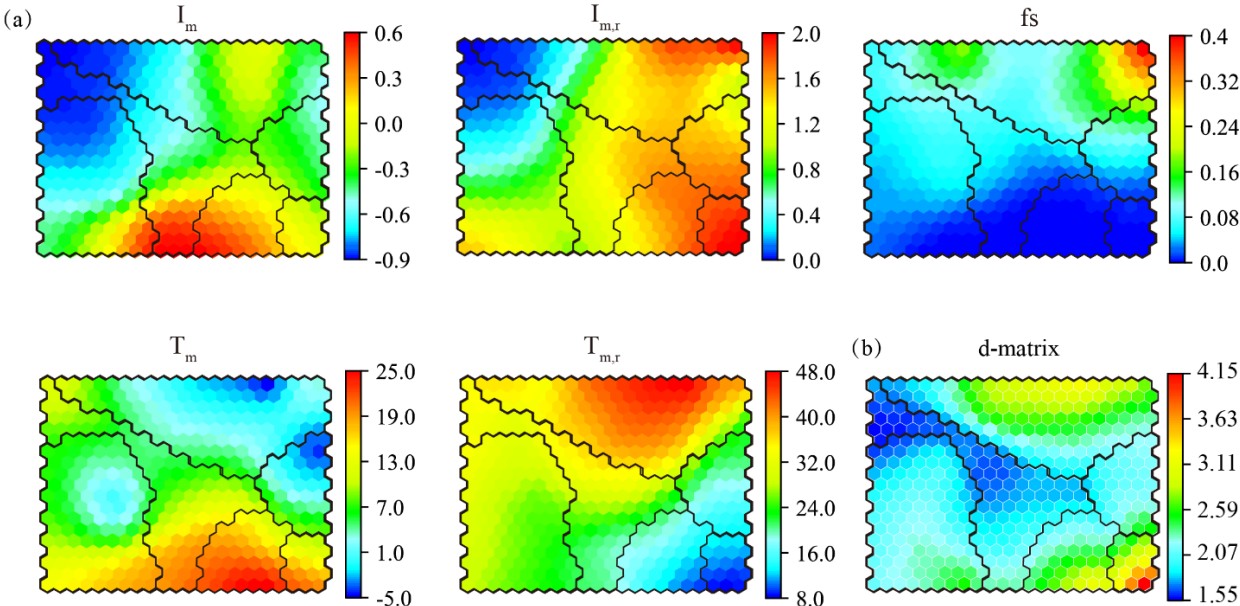

**Figure 2.** Results of SOM clustering: (a) component planes showing weight vector values of climate variables, and (b) clustering distance matrix (d-matrix).

### 3.1.3 FCM clustering results

Before clustering the output neurons from the SOM competitive layer using the FCM algorithm, two validation metrics were calculated (i.e., $DBI$ and $SC$) to determine the optimal number of clusters. An experiment was conducted to test the number of clusters from two to the maximum number determined by the AP algorithm. The optimal number of clusters was theoretically determined by minimizing $DBI$ and maximizing $SC$. They showed the best values when the number was up to six, and then $DBI$ increased with an increase in the number of clusters, and $SC$ changed slowly. Therefore, we defined six representative climates in continuous climate index space. The spatial distribution of these six climate groups in China is shown in Fig.3, where blue and red correspond to the low and high membership values belonging to a certain cluster, respectively. The results showed that there is a strong relationship between climatic clustering and mountain distribution on a large scale. The climate groups are described below in terms of their meteorological index characteristics.

Region 1 consists largely of Northwest China's desert regions, where the climate is extremely arid with little seasonal variation in the humidity index and some seasonal variation in temperature (Fig. 3a). Region 2 is located in southeastern China along the middle and lower reaches of the Yangtze River (Fig. 3b) and is characterized by a humid climate, high temperatures, and fairly steady humidity and temperatures throughout the year with little seasonal variation. Region 3 includes the Northeast Plain region of China (Fig. 3c), which is characterized by extremely cold temperatures, heavy snowfall, and wide seasonal temperature and precipitation variability. In the North China Plain region, region 4 occupies the middle and lower reaches of the Yellow River and exhibits more seasonal variation and higher temperatures than region 3 (Fig. 3d). Region 5 is located in





the basin north of the Tianshan Mountains (Fig. 3e) and is characterized by a high proportion of precipitation that is occupied

by snowfall, as well as a large seasonal variation in temperature and moisture content. Region 6 is located in the Tibetan

320 Plateau region (Fig. 3f), which is comparable to region 3 in terms of cold climate and high snowfall but with more consistent

regional temperatures and less seasonality.

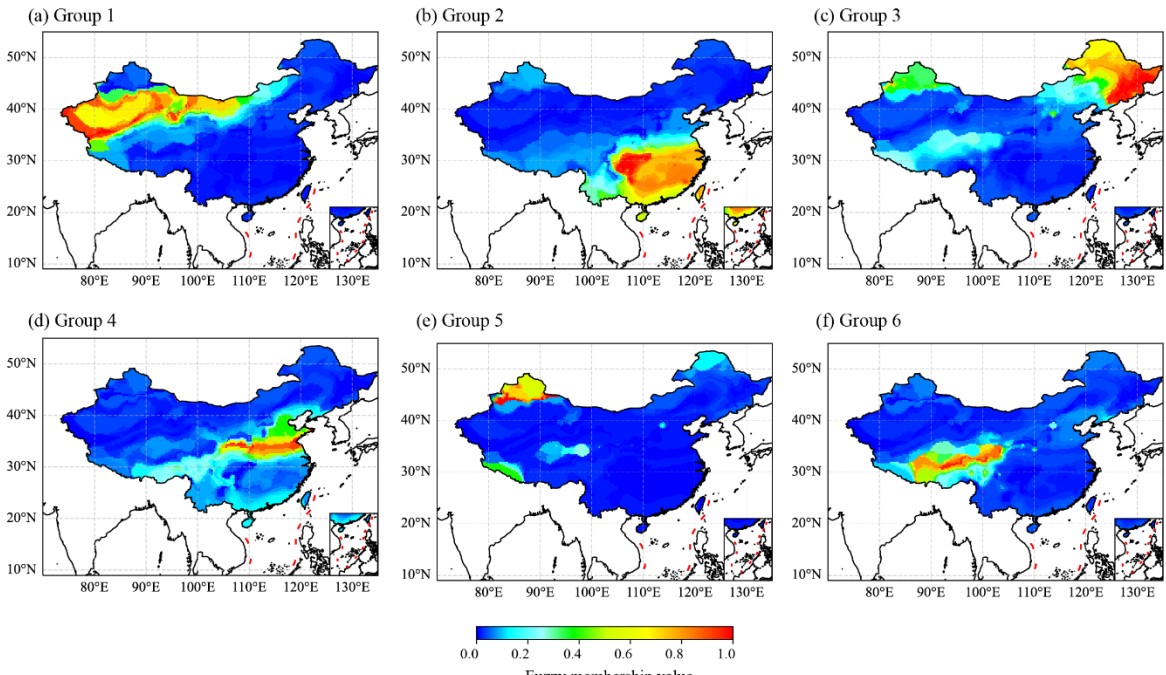

325 **Figure 3.** FCM clustering results across China. Colors represent fuzzy membership values for six climate regions.

The continuous spatial variation of the climate index causes the boundaries among different clusters to become blurred.

However, grid cells located far from the clustering boundaries tend to reflect a single climate mode with high membership

values. To facilitate catchment classifications under similar climate patterns, the climate type with the highest membership in

each grid cell was used as the dominant climate type for this cell (black borders in Fig. 2 represent the main clusters in the

330 component plane space). Regions 1,2,3 and 5 are on the extremes in the high-dimensional space of the climate index, and their

climate features, characterized by extremes, can be roughly approximated (e.g., region 1 is always arid and low seasonal;

region 2 is always wet and no snow). Conversely, for non-extreme points in the high-dimensional space of the climate index

(regions 4 and 6), the characteristics are more complicated to define (Fig. S1).



## 3.2 Results of catchment classification

### 3.2.1 Correlations of catchment attributes in the different climate regions

The catchments in terms of climate homogeneity were classified based on the climate region map in this section. Initially, the catchment landscape characteristics were assessed in various climatic regions. It is hypothesized that climate differences affect the relationship between catchment attributes. To quantify this, Spearman's rank correlation coefficients between catchment attributes across climate regions were calculated. The differences between regional correlation coefficients and average correlation coefficients were then compared for catchment characteristics in China. According to Fig. 4, similar types of catchment feature indicators have high correlation for all climate regions ($\beta$ and $TI$, both belonging to topographic characteristics, correlation coefficient >0.9), whereas different types of features have low correlation ($\beta$ and $L$, which belong to topographic characteristics and topological characteristics, respectively, with a correlation coefficient <0.1). Additionally, catchment attribute correlations varied little between climatic regions (Fig. S2), and the Spearman rank correlation coefficient for most characteristics varies less than 0.4 in most cases (> 80%). Nonetheless, some catchment attributes differed by up to 0.8. The $NDVI$ and $H$ were the most noticeable. From a national perspective, vegetation and elevation are negatively correlated; as elevation increases, vegetation becomes sparser. There was a strong correlation between vegetation and elevation (r=0.7) in the highlands (region 6), while in the plains (regions 2 and 5), the direction of correlation shifted. This is because temperature often varies significantly with elevation in highland areas, and thus, vegetation growth is influenced by temperature gradient. In contrast, temperature in plain areas is more closely tied to latitude, and therefore, vegetation is less correlated to elevation. Considering that region 6 is located on the Tibetan Plateau and is characterized by complex topography, the correlation between part of the elevation and topographic features is inverse to that of the other five regions (for instance, elevation range and slope do not increase significantly as elevation increases). Although the correlation coefficient has some local differences across climate regions, they still have comparable relationships between catchment attributes.





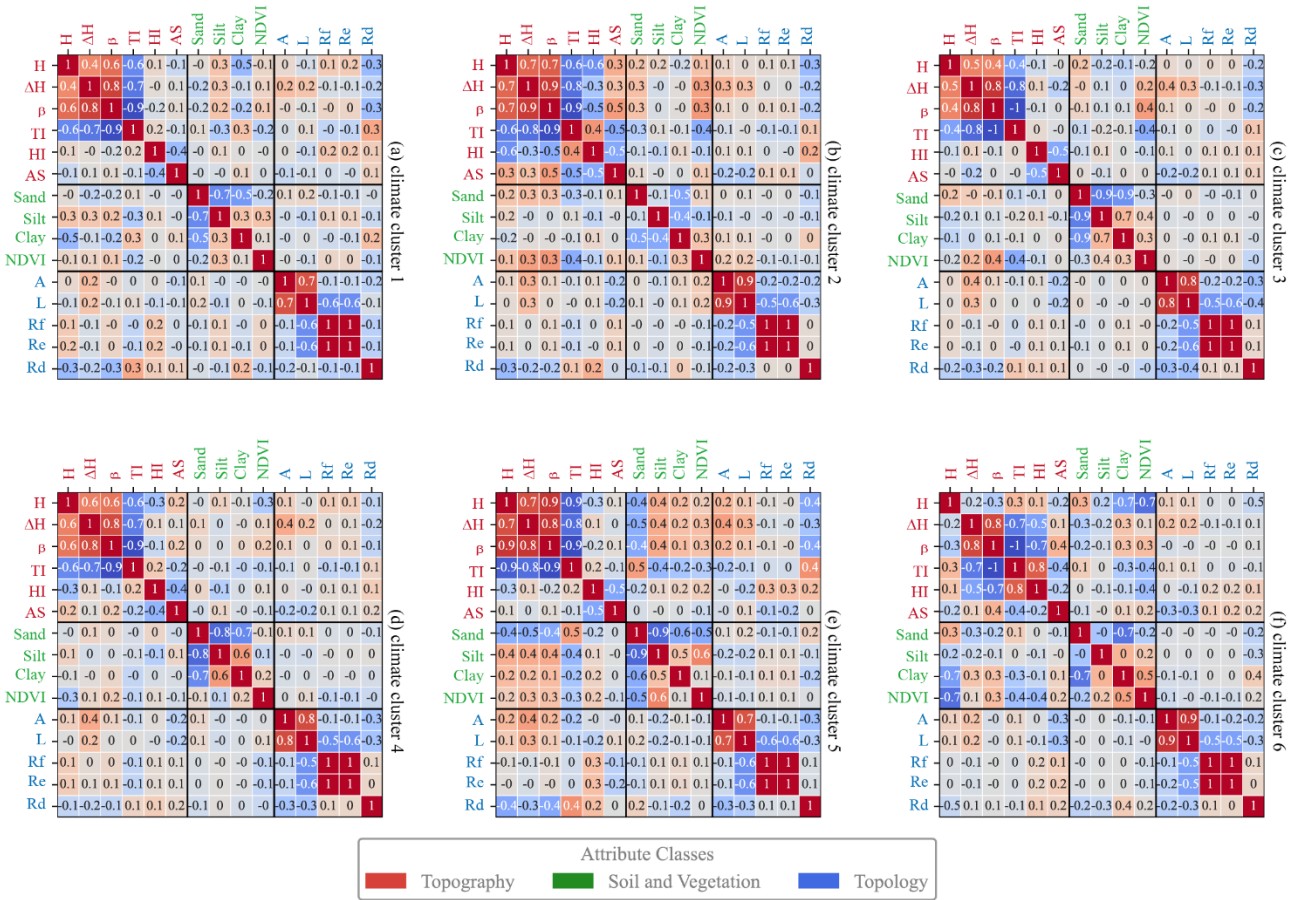

**Figure 4.** Spearman rank correlation coefficients for catchment attributes across different climate regions.

To eliminate redundancy, a principal component analysis was performed for each of the three types of catchment features to remove correlations between individual signatures of the same type. The results showed that catchments within the same climate region exhibited widely disparate topographic and topological characteristics in the principal component space (Fig. 5), implying the necessity for identifying similar catchments in the climate regions. Additionally, the soil and vegetation properties of the catchments in the same climate region were similar. While most catchments were characterized by a high percentage of sand and poor vegetation cover in regions 1 and 6, the catchments in region 2 had a high percentage of clay and dense vegetation cover. The remaining catchments fell between these two extremes. Each catchment feature type was analyzed using principal component analysis, and the principal component eigenvalues and proportions are shown in Table 2. Principal components were determined by eigenvalues greater than 1. As a result, two principal components were selected for each type of feature, which encompassed the most information in which the cumulative proportion exceeded 70%. In terms of topographic attributes, the first principal component had the highest correlation with the elevation and slope. The second most strongly correlated with *AS*; and the first principal component of soil and vegetation characteristics had the highest correlation

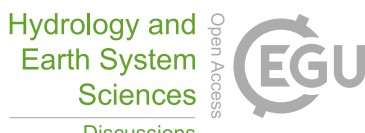

with sand content. The second correlated most strongly with *NDVI*; the first principal component of topological characteristics

370 had a strong correlation with *Re* and *Rf*, while the second had the highest correlation with river network density.

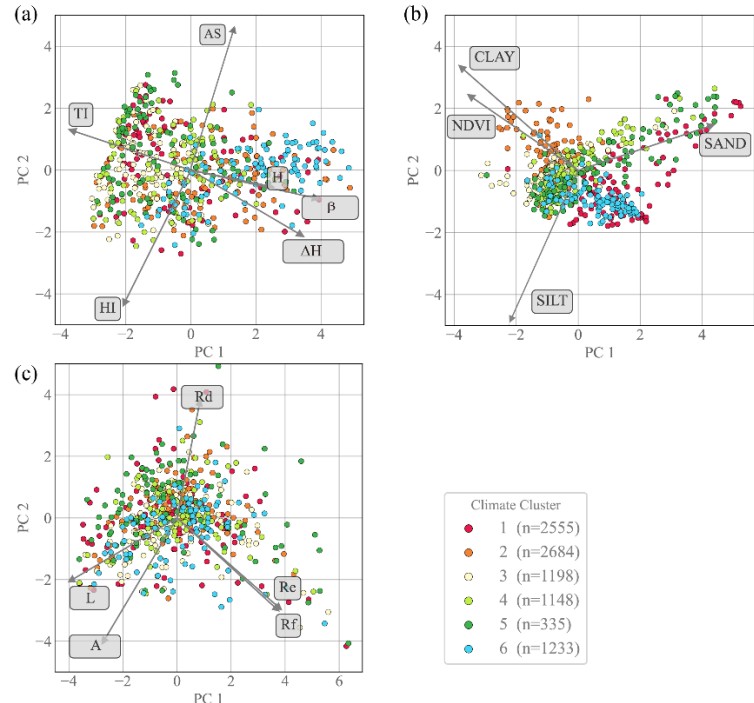

**Figure 5.** Biplots of the principal components: (a) topographic characteristics, (b) soil and vegetation, and (c) topological characteristics. Colors represent the climate clusters, while n indicates the number of catchments within each climate region.

**Table 2.** Eigenvalues and proportions of principal components for three types of landscape characteristics.

| Descriptor classes | Component | Eigenvalue | Proportion | Cumulative Proportion |
|---|---|---|---|---|
| Topographic characteristics | 1 | 3.24 | 54.02 | 54.02 |
| | 2 | 1.17 | 19.44 | 73.46 |
| | 3 | 0.68 | 11.37 | 84.83 |
| | 4 | 0.54 | 8.92 | 93.75 |
| | 5 | 0.31 | 5.14 | 98.89 |
| | 6 | 0.07 | 1.11 | 100.00 |
| Soil and vegetation characteristics | 1 | 2.31 | 57.71 | 57.71 |
| | 2 | 0.91 | 22.83 | 80.54 |
| | 3 | 0.54 | 13.43 | 93.98 |
| | 4 | 0.24 | 6.02 | 100.00 |
| Topological characteristics | 1 | 1.94 | 38.89 | 38.89 |
| | 2 | 1.71 | 34.22 | 73.11 |
| | 3 | 0.96 | 19.23 | 92.34 |
| | 4 | 0.26 | 5.15 | 97.49 |
| | 5 | 0.13 | 2.51 | 100.00 |





### 3.2.2 Catchment classification results using SOM-FCM algorithm

The SOM-FCM combined algorithm was used to cluster 13,487 catchments within six climate regions separately in China, and catchments were divided into 35 classifications. The membership values of the catchments within meteorologically homogeneous zones can be determined for various surface patterns. However, some catchment properties are spatially continuous, so catchments can belong to more than one mode. A membership threshold of 0.5 is generally interpreted as the cell belonging exclusively to its main cluster (Schwämmle and Jensen, 2010). Therefore, only membership values above 0.5 are regarded as belonging to a certain cluster, and below 0.5 are considered an indication that the catchment does not belong to a significant cluster and may have overlapping hydrological properties between multiple groups. The proportion of catchments with a single main cluster varied across meteorological regions. There were 44 percent of catchments in climate region 6 with insignificant main clusters and more complicated basin conditions, whereas 78 percent of catchments in meteorological region 5 had a delineated main cluster and low uncertainty in catchment mode. Catchment clusters are discussed in terms of their characteristics (Fig. S3) and location (Fig. 6); the crucial components of this description are presented in Table 3.

The catchments in climate region I were divided into seven classifications. Clusters 1 and 5 were mainly in the northern climate region 1 with relatively low elevation and gentle topography, defined by a high clay fraction and high hypsometric curve integral, respectively. Nevertheless, the soil texture in cluster 5 was coarser than that in cluster 1. Cluster 3 covered the majority of the west side of the climate region and was characterized by a high elevation range. This region showed high elevation and slope, low hypsometric curve integral, and low level of geomorphic development. The catchments in cluster 6 were in the Tarim Basin. They were characterized by a high sand fraction, flat terrain, low elevation, and sparse vegetation. Most of cluster 7 was located in the southern region. They possessed a relatively high level of geomorphic development, high silt fraction, and high elevation. Cluster 2, like cluster 4, was dispersed in space with a low drainage density. The two clusters had similar topography and soil characteristics, but the catchments in cluster 2 had larger areas and longer rivers.

There were five clusters in climate region II. Cluster 1 encompassed hilly areas in the southeast and southwest of climate region 2. The catchments were defined by high *NDVI* and had relatively low terrain and elevation, high clay fraction, and abundant vegetation. Cluster 2 consisted of catchments located in the Sichuan Basin and Lower Yangtze Coast. They were distinguished by a high gradient of hypsometric curves and have low elevation, moderate slope, and high silt fraction. Cluster 3, indicated by a moderate elevation range, demonstrated dispersed spatial distribution. The catchments here were very similar in their geographical characteristics to cluster 1, but cluster 3 shows a coarser soil texture. The majority of the catchments in cluster 4 were concentrated in the middle and lower reaches of the Yangtze River. The region had a gentle topography, with high hypsometric curve integra and low gradient of hypsometric curve, and the degree of landform development was modest. Cluster 5, characterized by high elevation, was mostly located in the Hengduan Mountains, and the catchments had high elevation and slope, as well as a high degree of geomorphic development.





The catchments in climate region III were grouped into six clusters. Cluster 1 was concentrated in the southern climatic region and was defined by a high topographic index. The catchments had an undulating topography, low slope degree, and coarse soil texture. Cluster 4 was identified in the Greater Khingan Range and Changbai Mountains, distinguished by a high silt fraction, relatively large slope degree, and low topographic index. Clusters 2 and 3 had a dispersed spatial distribution across the entire climate region. They were distinguished by low drainage density and high clay fraction, respectively, and both exhibit similar landscape characteristics. Cluster 2 had a slightly larger slope and elevation range, whereas cluster 3 had a rounded basin shape. The catchments in climate region IV were divided into six classifications. Clusters 1 and 5 were in the western Taihang Mountains and were distinguished by a low drainage density and high slope degree, respectively. These clusters had similarities in terms of relatively high slope, high elevation, and high geomorphic development, but cluster 1 showed slightly higher terrain undulation. Cluster 2 mainly covered the downstream area of the Yellow River, which was defined by a high hypsometric curve integral and has gentle topography with a low slope. Cluster 3 was situated to the east of the Taihang Mountains and had a low slope, low elevation, and low sand fraction with a high silt fraction. Clusters 4 and 6 were widely dispersed throughout the climate region. They were defined by a moderate gradient of hypsometric curves and low drainage density, respectively, and had similar topographic characteristics. However, cluster 4 had an overall higher sand fraction than cluster 6 did. Cluster 6 had larger areas and longer lengths than cluster 4.

The catchments in climate region V were grouped into 5 clusters. Cluster 3 was distributed in the Tianshan and Altay Mountain areas and was defined by a low drainage density. There are high altitudes, steep slopes, high silt, and dense vegetation in this area. The other clusters were distributed throughout the Junggar Basin. The landscape features of clusters 1, 2, and 5 were comparable and were defined by a high gradient of hypsometric curves, moderate drainage density, and high silt fraction, respectively. Cluster 1 had a slightly lower hypsometric curve integral and a lower degree of terrain undulation than clusters 2 and 5. Cluster 2 was characterized by a large area and longer river length than clusters 1 and 5. Cluster 4, characterized by a high topographic index, showed low topography and a moderate slope fraction with coarse soil and poor vegetation cover.

There were eight clusters in climate region VI. Overall, clusters 1 and 7 were distributed in the eastern part of the climate region and were defined by a high silt fraction and high *NDVI*, respectively. Clusters 2 and 6 were located in the southern part of the climatic region, which is defined by high *NDVI* and moderate drainage density, respectively. Cluster 2 had a finer soil texture, lower elevation, and was covered by a higher percentage of vegetation than cluster 6. Cluster 5 was situated in the northern climate region with a definition of high elevation, which has a relatively small degree of regional topographic relief and geomorphological development. Clusters 4 and 8 were defined by low drainage density, and the catchments in cluster 8 had narrow basin shapes with larger areas and longer lengths. As mentioned above, similar trends were observed for some clusters. Some similar catchments had spatial proximity, whereas others were far from each other.





**Table 3.** Properties of catchment clusters within climate regions. "Typical attribute" and "second attribute" refer to the attributes of the cluster with the lowest and second lowest coefficient of variation, which were scaled by the mean coefficient of variation of the dataset.

| Climate region | Cluster | Catchment numbers | Typical attribute | Second attribute |
|---|---|---|---|---|
| I | I-1 | 662 | High *Clay* | High *TI* |
| | I-2 | 568 | Low *Rd* | High *A* |
| | I-3 | 651 | high *ΔH* | High *H* |
| | I-4 | 361 | Low *Rd* | High *HI* |
| | I-5 | 365 | High *HI* | High *AS* |
| | I-6 | 605 | High *Sand* | High *AS* |
| | I-7 | 578 | High *Silt* | High *HI* |
| II | II-1 | 1110 | High *NDVI* | High *Clay* |
| | II-2 | 615 | High *AS* | Low *Clay* |
| | II-3 | 661 | Mid *ΔH* | Mid *Rd* |
| | II-4 | 690 | High *HI* | Mid *Rd* |
| | II-5 | 825 | High *H* | Mid *Sand* |
| III | III-1 | 218 | High *TI* | High *AS* |
| | III-2 | 403 | Low *Rd* | High *Clay* |
| | III-3 | 538 | High *Clay* | High *Silt* |
| | III-4 | 603 | High *Silt* | High *NDVI* |
| IV | IV-1 | 234 | Low *Rd* | High *ΔH* |
| | IV-2 | 211 | High *HI* | Mid *Rd* |
| | IV-3 | 256 | High *Silt* | High *TI* |
| | IV-4 | 221 | Mid *AS* | Mid *H* |
| | IV-5 | 388 | High *β* | High *H* |
| | IV-6 | 226 | Low *Rd* | High *A* |
| V | V-1 | 105 | High *AS* | Mid *Silt* |
| | V-2 | 69 | Mid *Rd* | High *A* |
| | V-3 | 129 | Low *Rd* | High *ΔH* |
| | V-4 | 93 | High *TI* | High *Rd* |
| | V-5 | 102 | High *Silt* | High *TI* |
| VI | VI-1 | 131 | High *Silt* | Mid *NDVI* |
| | VI-2 | 150 | High *NDVI* | Mid *Rf* |
| | VI-3 | 197 | Low *Rd* | High *H* |
| | VI-4 | 247 | High *H* | High *Sand* |
| | VI-5 | 317 | High *H* | High *TI* |
| | VI-6 | 143 | Mid *Rd* | High *β* |
| | VI-7 | 322 | High *NDVI* | High *Silt* |
| | VI-8 | 171 | Low *Rd* | Mid *Silt* |



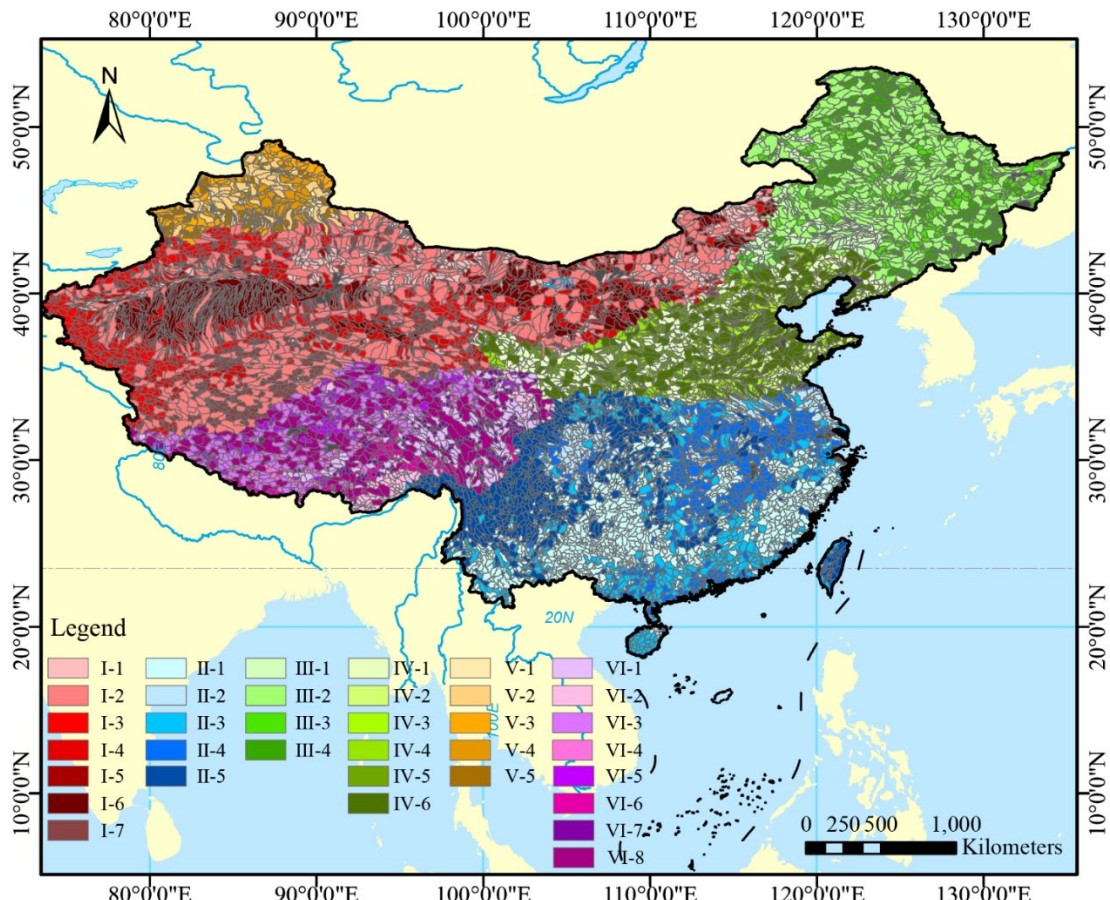

**Figure 6.** Map of the derived catchment classification across China.

### 3.3 Validation results for small catchments in China

Small catchments from various clusters were considered to validate the hydrological similarity with catchment classification.
The catchments selected for this study were based on the following criteria: (1) available hydrological data, (2) unregulated
catchments, and (3) containing a range of climate types and catchment classes(Li et al., 2018). Each catchment had 10–15
years of available daily continuous rainfall and runoff data with 10-35 flood events. The catchment areas range from 441 to
4,321 km$^2$, and the climate types span different regions containing various small catchment classes. The period of the data
record and small catchment classes for each catchment are listed, and the catchments are primarily located in climate regions
II and IV, which contain a variety of small basins (Table 4).






**Table 4.** Information and characteristics of the study catchments. Basin class indicates the class types of the subbasins in catchment.

| Station | River System | Area(km$^2$) | Number of flood events | Record period | Basin classes |
|---|---|---|---|---|---|
| Fenshuijiang | Qiantang River | 2619 | 10 | 2003-2014 | II-1 |
| Tunxi | Qiantang River | 2680 | 35 | 2008-2017 | II-1 |
| Chenhe | Yellow River | 1429 | 18 | 2003-2012 | II-5 |
| Daheba | Yangtze River | 2198 | 13 | 2013-2017 | II-5 |
| Banqiao | Yangtze River | 441 | 13 | 2000-2010 | IV-4 |
| Suide | Yellow River | 3897 | 22 | 2010-2017 | IV-4 |
| Daiying | Hai River | 4321 | 16 | 1990-2002 | IV-6 |
| Maduwang | Yellow River | 1605 | 12 | 2000-2010 | IV-6 |
| Dage | Hai River | 1864 | 17 | 1990-2008 | IV-4 |
| Zhidan | Yellow River | 779 | 15 | 2000-2010 | IV-1 |

The seasonal flow regime is a reliable predictor that can be used in comparative studies to analyze hydrological signatures across regions. Our research focused on the average seasonal pattern of runoff variability over the annual cycle in different regions, and variations in seasonal runoff between years were also discussed. Catchments located in different meteorological regions indicate clear regional heterogeneity (Fig. 7). The flow regime in climate region II presented multiple peaks following multiple peaks in precipitation in June and July during the same period. Monthly runoff was influenced by abundant

precipitation and has great interannual variability (e.g., the Fenshuijiang and Tunxi catchments). In climate region IV, the flow regime generally had a single peak after July, and the runoff showed a noticeable lag in peak monthly compared to monthly precipitation. The catchments had small runoff and low inter-annual variability (e.g., Daiying and Dage catchments). Catchments in the same climate region had comparable seasonal flow regimes connected to climate patterns rather than catchment landscapes. This is because seasonal runoff variations are directly driven by the relative seasonality of precipitation

and potential evaporation, whereas the landscape characteristics of catchments tend to influence runoff indirectly by affecting soil and groundwater storage.



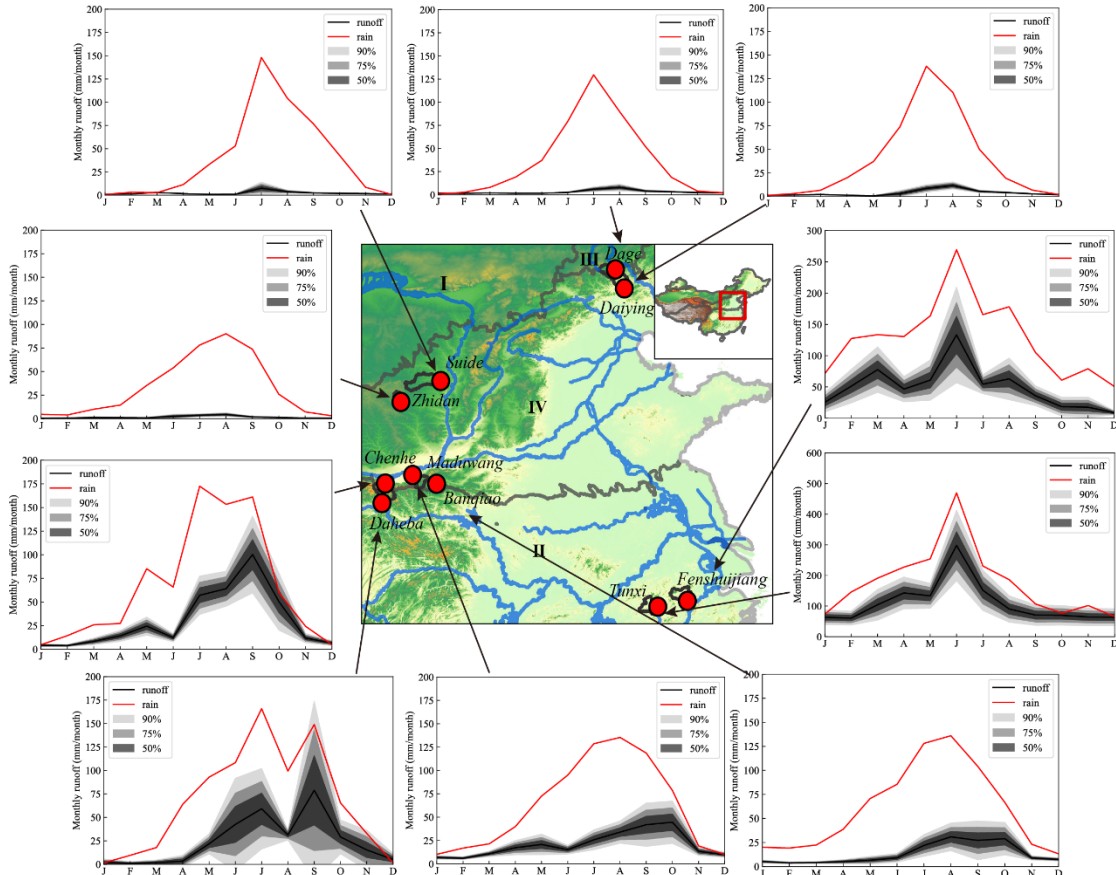

**Figure 7.** Regional differences in seasonality of precipitation and runoff.

The flow duration curve (FDC) is a valuable tool for diagnosing rainfall-runoff responses in gauged catchments at a holistic
level. The hourly FDC in flood event periods from different catchments located across several cluster regions showed
significant differences in the runoff regime (Fig. 8), illustrating the classification of the 10 catchments into five homogeneous
regions: (1) Tunxi, Fenshuijiang, (2) Chenhe, Daheba, (3) Daiying, Maduwang, (4) Dage, Suide, Banqiao, and (5) Zhidan.
Among them, the Tunxi and Fenshuijiang catchments belong to II-1, which has low elevation, fine soil, and dense vegetation.
These areas experience relatively stable humid climates with abundant precipitation, resulting in similar surface runoff and
stable groundwater flow processes.. The FDC curves represent a gentle flow, with high flow values obtained in the upper
portion, and stable and gentle flow in the lower flow portion. The Chenhe River and Daheba catchments belong to category
II-5, characterized by relatively high elevations and steep terrain. These areas experience significant high flows during
precipitation periods, with the flow becoming more gradual as the exceedance percentage increases, indicating higher base
flow levels. Zhidan is located in IV-1, which is distinguished by a hazardous terrain with a high degree of geomorphic
development, high elevation, and coarse soils. This climate is reflected in seasonal climate changes and low precipitation. The
FDC in Zhidan indicated a steeper curve with rapid regional surface runoff as the major flow, but essentially no subsurface





runoff activity. FDCs in the other catchments showed a situation between these two states. These results indicate that catchment classification based on climate and landscape can reflect the performance of runoff characteristics and differences.

Overall, climate appears to be the most important factor for medium- or long-term hydrological signatures, such as seasonal
runoff, whereas landscape features are more essential to hydrological features at the flood event scale. The results and comparison show that the hydrological similarity of the catchment is more difficult to identify than the spatial proximity. Using spatial proximity may prove effective only in areas where hydrological behavior gradually changes. However, in regions with high spatial variability, spatial proximity cannot be used to establish hydrological similarities (Knoben et al., 2018). Additionally, catchments with large spatial distances are capable of exhibiting similar hydrological characteristics (Maduwang
and Daiying, year). Hydrological complexity is reflected in various climates and individual signatures. The combined indicators appear to depict a dynamic that is climatic in origin, but are influenced by catchment characteristics (Berghuijs et al., 2014b; Jehn et al., 2020). The validation experiment demonstrated that the hydrological similarity of catchments at different scales can be reflected through classification based on climate and landscape characteristics.

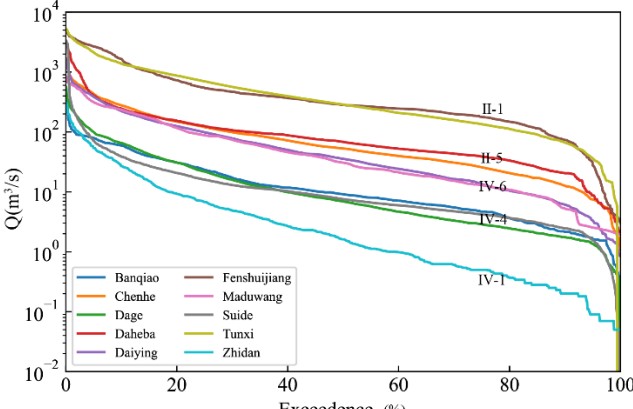

Figure 8. Flow duration curves for the test catchments.

## 4 Discussion

### 4.1 The necessity and boundary effects of dividing meteorologically homogeneous zones

Dividing small catchments based on meteorologically homogeneous zones involves grouping regions with similar climatic characteristics to understand hydrological processes. This approach is based on the premise that climate, as the primary driver
of hydrological behavior, significantly influences water resource availability, flow patterns, and overall watershed dynamics (Kuentz et al., 2017; Zhang et al., 2022). We believe that climatic variables such as precipitation and temperature play a critical role in shaping hydrological processes. Differences in climatic conditions lead to significant variations in the hydrological responses of watersheds (Addor et al., 2018; West et al., 2022; Wu et al., 2020). Studies have demonstrated that regionalization techniques are more effective when applied to areas with similar climatic conditions (Bharath and Srinivas, 2015; Hazarika





and Sarma, 2021; Samantaray et al., 2021).

Classifying small catchments with similar climatic conditions enables the development of more accurate regional hydrological models, thereby improving water resource management and planning (Liu et al., 2021; Pagliero et al., 2019).

However, the application of meteorologically homogeneous zones encounters boundary effect issues in practice. This issue arises because climatic variables, such as temperature and precipitation, generally change gradually across space. The introduction of artificial boundaries can result in inaccurate classifications, particularly at the edges of these zones. Although climate change is continuous (Viviroli et al., 2011), creating abrupt boundaries can disrupt natural gradients, potentially leading to errors in hydrological modelling. An effective solution to this issue is the use of fuzzy clustering techniques, which allow each catchment to belong to multiple clusters with varying degrees of membership (Bharath and Srinivas, 2015; Sreeparvathy and Srinivas, 2022). Some studies have introduced transition zones or buffer areas between clusters to create gradients in climatic and hydrological characteristics, thereby reducing the impact of abrupt boundaries (Cantidio and Souza, 2019; Wen et al., 2017). Hierarchical clustering methods also offer a multi-scale approach, starting with broader climatic zones and then making more detailed subdivisions, thus maintaining the continuity of climatic variables at various spatial resolutions (Sreeparvathy and Srinivas, 2022). Finally, fuzzy clustering was adopted to preserve the continuity of climatic variables, providing a more accurate representation of hydrological regions. This approach resulted in well-classified similar catchments.

## 4.2 The complexity of catchment hydrological behavior

Watersheds are complex systems resulting from the interaction of climatic and landscape processes, leading to the co-evolution of hydrological processes (Addor et al., 2018). Therefore, identifying the primary drivers of changes in hydrological response is challenging (Jehn et al., 2020). This watershed study encompasses mountainous, plateau, and plain regions but does not account for the impact of human activities. Due to data limitations, hydrological characteristics were not used as a zoning factor; instead, the study was based on a fundamental assumption: watersheds with similar climatic and basin characteristics exhibit comparable hydrological behavior (Dallaire et al., 2019; Jehn et al., 2020). This assumption is based on the premise that hydrological behavior in watersheds is largely influenced by climatic factors (e.g., precipitation and evaporation) and basin characteristics (e.g., topography, soil type, and land use). For example, Jehn et al. (2020) pointed out that watersheds under different climatic conditions typically exhibit different hydrological responses because precipitation and evaporation patterns directly determine the inputs and outputs of the hydrological cycle. Additionally, topography and soil type affect water infiltration and runoff pathways, thereby influencing the hydrological response of the watershed (Kuentz et al., 2017). On the other hand, clustering analyses using catchment attributes and flow signatures showed significant consistency in clustering results, further demonstrating the significant control catchment attributes have on hydrological behavior (Du et al., 2023).

However, this assumption is not without limitations. First, the hydrological behavior of a watershed is not determined solely by external factors like climate and topography, but also by complex interactions within internal hydrological processes (Mcdonnell et al., 2007). Co-evolutionary processes within a watershed, such as dynamic changes in vegetation, soil, and





geomorphology, can lead to watersheds with similar climatic and topographical features exhibiting different hydrological behaviors (Bogaart et al., 2016). Second, anthropogenic factors such as land-use changes and water resource management practices can significantly alter a watershed's hydrological behavior (Dwarakish and Ganasri, 2015). Human activities, such

as urbanization and agriculture, can exert a greater influence on a watershed's hydrological processes than natural characteristics (Neupane and Kumar, 2015). Consequently, even watersheds with similar climatic and topographical features may exhibit significantly different hydrological behaviors due to human interference. Therefore, watershed hydrological behavior results from the interaction of multiple factors, necessitating a comprehensive analysis to accurately understand watershed similarities and predict hydrological responses.

**5 Conclusions**

Hydrologically homogeneous catchments were clustered in China based using both SOM and FCM algorithms. As runoff is the result of the interaction between climate and catchment processes, an index system for climate and landscape characteristics was constructed. We assembled six climatic regions and 35 watershed classifications that fully reflect the regional hydrological characteristics of China. Furthermore, 10 catchments belonging to different classifications were selected to verify and analyze

homogeneous regions. The results indicated that hydrological behavior is better characterized through climate and landscape characteristics in catchments. Moreover, climate-homogeneous regions respond to hydrological behaviors at medium- or long-time scales, whereas catchment classification regulates hydrological processes at the flood event scale. Combining the SOM and FCM algorithms provides a comprehensive quantitative evaluation of complex catchment structures. SOM enables complex, high-dimensional input data to be converted into intuitive 2D output surfaces. FCM utilizes membership values to

address identifying catchments with fuzzy boundaries. There is no particular classification for one catchment that allows greater flexibility in the selection of a catchment for comparative studies or parameter transplantation in ungauged catchments.

The issue of flood simulation and forecasting in ungauged catchments has been a challenge owing to the lack of effective observational data. The development of a better hydrological similarity classification is critical for transferring hydrological model parameters and runoff simulations. In a high-dimensional heterogeneous feature space, the proposed method for

identifying similar catchments at the basin scale can provide a guide for selecting similar basins. However, the hydrological behavior of catchments is not only determined by external characteristics such as climate and topography but also influenced by the complex interactions of internal hydrological processes, making it challenging to provide an in-depth description of catchment hydrological characteristics. Future research should focus on finer scales and consider including additional hydrological features, taking into account the impact of scale and feature selection on hydrological classification. Open data

sources enable new regionalization studies, showing great potential for generating new knowledge and hydrological insights across various environmental conditions. For the Chinese region, however, there is a lack of homogeneous datasets on runoff characteristics and human impacts. Therefore, it is crucial to make more public sector data available and to construct



standardized datasets for research purposes. This is of great importance for understanding the causes of catchment hydrological processes and for conducting regional studies.

*Data availability.* The observation-driven datasets analyzed in this study are publicly available as referenced within the article. Meteorological and land surface products datasets utilized in this study can be accessed through the following sources: Precipitation and Temperature datasets (http://data.cma.cn/), Potential Evapotranspiration dataset (https://www.ceda.ac.uk/), watershed boundary HydroSHEDS dataset (https://www.hydrosheds.org/page/overview), ASTER GDEMV2 digital elevation model (http://www.gscloud.cn), Soil and Vegetation characteristics (http://www.issas.ac.cn), Spot/vegetation NDVI dataset 580 (https://www.resdc.cn). The self-organizing map clustering methodology used in this study is available online (https://github.com/sevamoo/SOMPY). Python code for data computation, analysis, and graphical visualization can be obtained from the respective authors upon reasonable request.

*Competing interests.* The authors declare that they have no conflict of interest.

*Acknowledgments.* This study was supported by the National Key Research and Development Program of China 585 (2023YFC3006505), Fundamental Research Funds for the Central Universities of China (B240203007), and the fund of National Key Laboratory of Water Disaster Prevention (524015222).








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
