# Peer review of "Integrated Catchment Classification Across China Based on Hydroclimatological and Geomorphological Similarities Using Self-Organizing Maps and Fuzzy C-Means Clustering for Hydrological Modeling"

_Hydrology and Earth System Sciences, 2024_

## Author Comment (AC1)

**General comments**

**Comments 1: Table 3**    Why are the attributes in Table 3 selected based on the coefficient of variation?

**Response 1:** Thank you for your valuable comments on our research. The attributes in Table 3 were selected based on the coefficient of variation (CV) as it serves as a measure of relative variability, which is particularly useful in identifying the most stable attributes within each catchment cluster. By focusing on the attributes with the lowest and second lowest CV, we aimed to highlight those that exhibit consistent behavior across different catchments within each climate region, making them more representative of the typical characteristics of the clusters. Additionally, using CV allows us to account for the inherent variability of the dataset, thereby ensuring that the selected attributes are robust and not unduly influenced by outliers or extreme values. Scaling these attributes by the mean CV of the dataset further normalizes the variability, providing a clearer comparison between the clusters.

To enhance the clarity of the manuscript, we will add a detailed explanation in the text to further justify the use of the coefficient of variation (CV) for describing the attributes of catchment clusters. Once again, thank you for your valuable feedback, which has helped us improve the clarity of our work.

**Comments 2:  Figure 2**   It is suggested to include an explanation of the d-matrices in the methodology. Consider moving the statement "Vesanto (1999) suggested that SOM results can be expressed in the form of two types…" from L279 to section 2.1.2 and expand on it in more detail.

**Response 2:** Thank you for your constructive suggestion.  We agree that an explanation of the d-matrices in the methodology would improve the clarity of the manuscript.  We will expand on this concept by providing a more detailed description of the d-matrices in Section 2.1.2, where we will explain their role and how they are derived in the context of the self-organizing map (SOM) methodology.

Additionally, we will move the statement "Vesanto (1999) suggested that SOM results can be expressed in the form of two types⋯" from Line 279 to Section 2.1.2, as you suggested.  This will allow us to elaborate further on this aspect and provide a clearer explanation of how the results are represented in the SOM framework.

Thank you for your valuable feedback, which has helped us improve the clarity and rigor of our work.

**Comments 3: Figure 3** Consider adjusting the color band so that the color corresponding to 0.5 is set to white. This would better highlight basins belonging to a cluster with higher confidence.

**Response 3:** Thank you for your valuable suggestion. We agree that adjusting the color band will improve the clarity of the distribution in Figure 3, particularly in better highlighting the basins with higher confidence that belong to the clusters. We will adjust the color band corresponding to membership values less than 0.5 to white, ensuring that the visual representation more effectively distinguishes basins with different levels of confidence. This adjustment will further enhance the interpretability of the results.

**Comments 4: Figure 6** It is recommended to use different color schemes for the third and fourth categories, as their current colors are too similar and not effective.

**Response 4:** Thank you for your helpful suggestion. We agree that the current color scheme for the third and fourth categories in Figure 6 is too similar and may cause confusion. We will modify the color scheme to select more distinct colors for these two categories, ensuring a clearer visual distinction between them. This change will help enhance the figure's effectiveness in presenting the data.

**Comments 5: Figure 7** Consider clearly marking the boundaries of each climate zone in the figure and labeling the basin class in the subplots.

**Response 5:** Thank you for your valuable suggestion regarding Figure 7. In the original version, we used gray solid lines to delineate climate zone boundaries and labeled the climate regions. To address your feedback, we will enhance the figure by using more distinct colors and labels to clearly highlight the boundaries of each climate zone. Additionally, we will label the basin classes in the subplots. These improvements will help enhance the readability and clarity of the figure. We appreciate your constructive input, which has strengthened the clarity of our visual presentation.

**Comments 6: Introduction** The paper overlooks previous catchment classification studies conducted in China:

Luo, K. (1954) Draft of natural geography regionalization of China. (in Chinese)

罗开富,1954. 中国水文区划草案.

Xiong, Y., Zhang, J., et al. (1995) Hydrology Regionalization of China, Science Press, Beijing.(in Chinese)

熊怡,张家桢,等,1995. 中国水文区划. 科学出版社

Liu, C., Zhou, C., et al. (2014) Chinese Hydrological Geography, Science Press, Beijing

刘昌明，周成虎等，2014. 中国水文地理. 科学出版社

Xu, H., Wang, H., Liu, P. (2024). Identifying control factors of hydrological behavior through catchment classification in Mainland of China. Journal of Hydrology, 645, 132206. DOI: 10.1016/j.jhydrol.2024.132206

**Response 6:** Thank you for pointing out these important references. We appreciate your suggestion to include previous catchment classification studies conducted in China. We acknowledge the valuable contributions of Luo (1954), Xiong et al. (1995), Liu et al. (2014), and Xu et al. (2024) in this field. In response, we will expand the introduction after Line 66 to include a discussion of these studies and their relevance to our work. This will provide a more comprehensive background for our research and highlight how our approach builds on and complements the existing studies in China.

**Comments 7: Methodology** In L180, the author claims FCM has "low sensitivity to initialization." I am curious if this is the case, and it might be beneficial to demonstrate FCM results under multiple initializations.

**Response 7:** Thank you for raising this important point regarding the sensitivity of Fuzzy C-Means (FCM) clustering to initialization. While FCM is generally considered less sensitive to initialization compared to hard clustering methods like K-Means, we acknowledge that initialization can still influence the results, particularly in complex, high-dimensional datasets. To address this concern and strengthen the robustness of our findings, we propose the following actions:

● **Clarification of the Statement:**

We will revise the statement in Line 180 to more accurately reflect the behavior of FCM. Instead of claiming that FCM has "low sensitivity to initialization," we will state that FCM is "relatively less sensitive to initialization compared to hard clustering methods, but initialization can still affect the results, particularly in high-dimensional datasets."

- **Demonstration of FCM Results Under Multiple Initializations:**

To empirically demonstrate the impact of initialization on FCM results, we will conduct additional experiments with multiple initializations. Specifically, we will run the FCM algorithm 10 – 20 times with different random initializations and compare the resulting cluster memberships and centroids.

We will include a brief analysis of the variability in cluster results across initializations, such as the average difference in membership values or the stability of cluster centroids. This will provide quantitative evidence of the sensitivity (or lack thereof) of FCM to initialization in our specific application.

By addressing this comment, we aim to provide a more rigorous and transparent analysis of the FCM algorithm's behavior in our study. Thank you for your insightful feedback, which has helped us improve the methodological robustness of our work.

**Comments 8: Methodology** It is suggested that the methods used in the results section be introduced in the methodology, highlighting the logic and approach rather than just detailing the SOM and FCM algorithms. A flowchart would be helpful if possible.

**Response 8:** Thank you for your valuable suggestion. We agree that the methodology section should provide a clearer and more comprehensive explanation of the overall logic and approach used in the study, rather than focusing solely on the technical details of the Self-Organizing Maps (SOM) and Fuzzy C-Means (FCM) algorithms. To address this, we will revise the methodology section to include the following improvements:

- **Enhanced Explanation of Logic and Approach:**

We have introduced a dedicated section at the beginning of the methodology to outline the overall workflow and rationale for integrating Self-Organizing Maps (SOM) and Fuzzy C-Means (FCM) clustering for catchment classification. To further enhance clarity, we will expand this section to include a discussion of why these methods were chosen, how they complement each other, and how they address the challenges of classifying catchments across diverse hydroclimatological and geomorphological conditions.

Additionally, we will explicitly state the steps involved in the process, such as data preprocessing, variable selection, dimensionality reduction, clustering, and validation, to provide a logical flow of the methodology.

● **Addition of a Flowchart:**

A flowchart will be added to visually summarize the methodological steps, from data collection and preprocessing to the final classification and validation. This will help readers better understand the sequence of operations and the relationships between different components of the methodology.

By addressing this comment, we aim to provide a more rigorous and transparent explanation of our methodology. Thank you for your insightful feedback, which has helped us improve the clarity and logical flow of our work.

**Comments 9: Methodology** How to classify catchment from climate region to basin class? FCM? If so, are the inputs to FCM the features in Table 1 or their principal components?

**Response 9:** Thank you for your question regarding the classification process from climate regions to basin classes. We appreciate the opportunity to clarify the methodology and ensure it accurately reflects the steps taken in the study. Below, we provide a detailed explanation of the two-step classification process used in our research.

● **The classification process involves two main steps:**

Step 1: Climate Region Classification

The SOM-FCM algorithm is first applied to classify catchments into climate regions. In this step, the inputs to the SOM algorithm are the climate indices derived from the data preparation process described in Section 2.1.1 (Selection of Climate Indices). These indices capture key meteorological characteristics, such as aridity, temperature, and precipitation patterns. The SOM algorithm reduces the high-dimensional climate data into a 2D map of neurons, which is then clustered using FCM to define distinct climate regions. This step ensures that catchments within each climate region share similar meteorological properties.

Step 2: Basin Classification within Climate Regions

Within each climate region, the SOM-FCM algorithm is applied again to classify catchments into basin classes. Here, the inputs to the SOM algorithm are the principal components of the catchment descriptors (derived from the original features listed in Table 1). These principal components reduce the dimensionality of the data while retaining the most important variability in both climate and geomorphological

characteristics. The output neurons (weight vectors) from the SOM algorithm, which represent the reduced-dimensional representation of the original features, are then clustered using FCM to define basin classes. This step ensures that catchments within each basin class share similar hydrometeorological and geomorphological characteristics.

- **Rationale for Using Principal Components**

Principal component analysis (PCA) is employed to address multicollinearity among the input variables and improve computational efficiency. By transforming the original features into principal components, we reduce redundancy while preserving the essential information needed for classification. The SOM algorithm further processes these principal components into a 2D representation, which is then used as input to FCM for clustering. This approach ensures a robust classification that captures both broad-scale climate patterns and fine-scale geomorphological variations.

To address this comment and accurately reflect the methodology, we will revise the methodology section to explicitly describe the two-step SOM-FCM process and clarify the inputs to each step. By doing so, we aim to provide a clearer and more accurate explanation of the classification process. Thank you for your insightful feedback, which has helped us improve the methodological rigor and clarity of our work.

**Comments 10: Results** Were the selected 10 small watersheds affected by human activities, such as agricultural water use or urban consumption? Would this impact the results?

Using 10 small watersheds for validation might be insufficient. If the author is willing, more runoff data can be found in NESSDC (https://www.geodata.cn), such as:

DOI: 10.12041/geodata.30184613892738.ver1.db

DOI: 10.12041/geodata.69811525443157.ver1.db

DOI: 10.12041/geodata.31258482188424.ver1.db

**Response 10:** Thank you for your insightful comments regarding the potential impact of human activities on the selected small watersheds and the valuable suggestion to expand the validation dataset. We appreciate your feedback, which has helped us identify areas for improvement in our study. Below, we address your concerns and outline the steps we will take to enhance the robustness of our results.

- **Impact of Human Activities on Selected Watersheds:**

The 10 small watersheds selected for validation were chosen based on the criteria of being unregulated catchments with minimal human interference, as stated in Section 3.3. This selection was made to ensure that the hydrological behavior of these catchments primarily reflects natural processes rather than anthropogenic influences. However, we acknowledge that even in unregulated catchments, low-level human activities (e.g., agricultural water use or small-scale urban consumption) might still exist and could potentially impact the results. To address this, we will conduct a detailed analysis of land use data for each catchment to assess the extent of human influence. This analysis will be included in the revised manuscript, and any catchments showing significant human impact will be either excluded or discussed in the context of their potential effects on hydrological signatures. We agree that future studies should incorporate more detailed assessments of anthropogenic impacts to further improve the reliability of catchment classification.

- **Expanding the Validation Dataset:**

We greatly appreciate your suggestion to use additional runoff data from the National Earth System Science Data Center (NESSDC). We will incorporate the datasets you recommended (DOIs: 10.12041/geodata.30184613892738.ver1.db, 10.12041/geodata.69811525443157.ver1.db, and 10.12041/geodata.31258482188424.ver1.db) to expand our validation dataset. By including more catchments, We will be able to validate the classification approach across a wider range of hydrological conditions, ensuring its applicability to diverse environmental settings, while also assessing the generalizability of the results to other regions in China, particularly those with varying climate and landscape characteristics. Thank you for your valuable feedback, which has significantly improved the quality of our work.

**Comments 11: Discussion** The discussion needs to emphasize the connection with the results. Currently, the discussion section seems to introduce existing knowledge within the basin. Perhaps discussing similarities and differences with similar studies, limitations, and potential applications would be more effective.

**Response 11:** Thank you for your valuable feedback regarding the discussion section. We agree that the discussion should more effectively emphasize the connection with

the results, highlight similarities and differences with similar studies, and address the limitations and potential applications of our work. Below, we outline the revisions we will make to address these points.

● **Emphasizing the Connection with Results**

We will revise the discussion to explicitly connect our findings to the validation results. we will discuss how the seasonal flow regimes and flow duration curves (FDCs) observed in the 10 small catchments (and the expanded dataset) support the effectiveness of our classification approach. We will also highlight how the differences in hydrological behavior between catchments in different climate regions align with the classification results and provide insights into the underlying hydrological processes.

● **Comparing with Similar Studies**

We have explored in the discussion section that climate and watershed characteristics can exhibit similar hydrological behavior, and we will build on this to compare with similar studies using other clustering methods for catchment classification. We will also highlight the unique aspects of our approach, such as the integration of SOM and FCM and the use of both climate and geomorphological characteristics, and how these contribute to the improved classification of catchments.

● **Highlighting Potential Applications and Limitations**

We will expand the discussion to highlight the potential applications of our classification approach. we will discuss how our method can be used to improve hydrological modeling in ungauged catchments, support water resource management, and inform climate change adaptation strategies. We will add a dedicated subsection to discuss the limitations of our study. We will acknowledge the potential impact of human activities on the selected catchments and the challenges of generalizing our results to regions with significant anthropogenic influences.

Thank you for your valuable feedback, which has significantly improved the quality of our work.

**Comments 12: Discussion**   The features used in this paper do not consider any human activities. How might this affect the results of catchment classification? Given the significant human activities in many regions of China, how should we interpret or use the classification results obtained without considering human activities?

**Response 12:** Thank you for your valuable feedback. The features used in our study (climate indices and geomorphological characteristics) do not explicitly account for human activities. While this allows us to focus on natural hydrological processes, it may limit the applicability of our classification results in regions where human activities significantly alter catchment behavior. We will add a discussion of how human activities might affect the results of catchment classification. We discuss how in areas with significant agricultural or urban development, human activities can outpace natural hydrological processes, leading to deviations from the patterns identified in our study.

we will suggest that our classification results are most applicable in regions with minimal human impact, such as remote or protected areas. For regions with significant human activities, we will recommend combining our classification framework with additional data on land use, water management practices, and other anthropogenic factors to improve the accuracy of hydrological modeling and predictions.

We will further refine this part to ensure that our research contributions and innovations are clearly explained. Thank you again for your feedback, and we will make the necessary improvements based on your suggestions.

**Comments 13: L460-471** This part is not easy to understand. Especially, I didn't understand this sentence: L464 "The flow regime in climate region II presented multiple peaks following multiple peaks in precipitation in June and July during the same period."

**Response 13:** Thank you for your valuable feedback on our manuscript. We agree that this section could be more clearly written to improve readability and ensure that the findings are easily understood. we will provide a revised version of this paragraph, with a clearer explanation of the flow regime patterns in climate region and their relationship to precipitation.

Thank you again for your feedback, and we will make the necessary improvements based on your suggestions.

**Comments 14: L495-498** What do "combined indicators" refer to? What does "at different scales" mean? Basin area? Time?

**Response 14:** Thank you for your question regarding the terms "combined indicators" and "at different scales" in Lines 495–498. We agree that these terms require

clarification to ensure readers fully understand their meaning and significance within the context of our study.

We will clarify that "combined indicators" refer to the integration of climate indices (e.g., moisture index, temperature, snow fraction) and geomorphological characteristics (e.g., elevation, slope, soil texture) used in our classification framework. These combined indicators capture both climatic and landscape factors that influence hydrological behavior. To avoid ambiguity, we will revise the relevant terms to make their meaning clearer.

Additionally, we will clarify that "at different scales" refers to both spatial scales (e.g., basin area, regional and local patterns) and temporal scales (e.g., seasonal runoff, flood events). This reflects the multi-scale nature of hydrological processes, which are influenced by both large-scale climate patterns and small-scale landscape features. Specifically, climate patterns primarily influence seasonal runoff (medium- to long-term scales), while landscape characteristics play a more significant role in flood events (short-term scales). To eliminate ambiguity, we will explicitly use the terms "spatial scale" and "temporal scale" in the revised manuscript.

Once again, thank you for your valuable feedback, which has helped us improve the clarity of our work. We will make the necessary adjustments in the revised manuscript.

**Comments 15: L560-561** What does "There is no particular classification for one catchment that allows greater flexibility in the selection of a catchment for comparative studies or parameter transplantation in ungauged catchments" mean?

**Response 15:** Thank you for your valuable feedback. we will provide a revised explanation to clarify the flexibility offered by our classification approach in selecting catchments for comparative studies or parameter transplantation in ungauged catchments.

The phrase "no particular classification for one catchment" is ambiguous and could be misinterpreted. It is intended to highlight that our classification framework does not rigidly assign a single classification to each catchment but instead allows for flexibility through the use of the SOM-FCM algorithm. we will include a brief explanation of how this flexibility benefits hydrological modeling and regionalization studies. The ability to identify catchments with overlapping characteristics supports more accurate parameter transplantation, as it accounts for the gradual transitions in hydrological

behavior between catchments. This approach is particularly valuable in regions with high spatial variability, where rigid classifications may fail to capture the complexity of hydrological processes.

We will further refine this part to ensure that our research are clearly explained. Thank you for your valuable feedback, which has helped us improve the clarity of our work.

**Comments 16: L556-557** The statement "Moreover, climate-homogeneous regions respond to hydrological behaviors at medium- or longtime scales, whereas catchment classification regulates hydrological processes at the flood event scale" needs to be strengthened in the results to support this conclusion.

**Response 16:** Thank you for your valuable comments. We agree that this conclusion needs stronger support from the results to ensure its validity and clarity. Below, we propose revisions to strengthen the connection between this statement and the results presented in the study.

We will revise the discussion to explicitly link this conclusion to the results presented in Section 3.3 (Validation Results for Small Catchments in China). Specifically, we will reference the seasonal flow regimes (Fig. 7) and flow duration curves (Fig. 8) to demonstrate how climate-homogeneous regions influence medium- to long-term hydrological behaviors and how catchment classification regulates flood event-scale processes. And we will also provide additional analysis or examples from the results to reinforce this conclusion.

We sincerely appreciate the reviewer's insightful suggestion, which has helped us improve the clarity and rigor of our work.

---

## Author Comment (AC3)

**General comments**

**Comments 1:** The novelty of the study is unclear to me. Is this the first study of fuzzy classification for small and medium-sized watersheds in China? Compared with other classified results, what are the major differences (not methodology) or improvements, such as hydrologic signatures? This should be elaborated in the Introduction and Discussion sections.

**Response 1:** Thank you for your valuable comments on our research. We appreciate your insight into the novelty of our study. To clarify, while there have been previous studies on catchment classification in China, such as those based on climate and geomorphological characteristics, our study contributes in several key areas.

First, our research specifically integrates Self-Organizing Maps (SOM) with Fuzzy C-Means (FCM) clustering for small and medium-sized catchments in China, which has not been done before. This hybrid approach allows us to better capture the fuzzy boundaries between catchments with similar hydrological behaviors, an important improvement compared to traditional methods that tend to assign catchments to distinct, rigid groups. This flexibility is particularly useful for applications in ungauged basins where hydrological data may be sparse.

Furthermore, there have been relatively few comprehensive explorations of catchment classification within China. Previous studies, such as Luo (1954), divided China into three grades and nine regions based on basin boundaries, flow patterns, and sediment content, marking the first hydrological zoning scheme for China. Xiong and Zhang (1995) further divided China into 11 regions based on the annual average runoff depth. Liu (2014) also proposed three regions based on topography and climate patterns. More recently, Huan Xu (2024) utilized Fuzzy C-Means (FCM) clustering and Classification and Regression Tree (CART) methods to extend catchment classifications to ungauged basins, dividing China's basins into five clusters.

In comparison to these existing classification methods, which often focus on discrete groupings or specific climatic zones, our approach provides a more nuanced classification. We first perform climate zoning and then conduct finer-scale classification based on geomorphological features within these homogenous climate regions. This dual-layer classification system accounts for the gradual transitions between catchments, making it more adaptable and detailed. By incorporating both

climate and geomorphological factors, our classification system offers a flexible framework that can be applied to ungauged catchments.

In summary, our study's novelty lies in its integrated approach, which combines SOM with FCM clustering for a multi-layered and more flexible classification of catchments. This innovative methodology provides a deeper understanding of hydrological behavior by accounting for both gradual transitions and the complex interaction between climatic and geomorphological factors, making it more suitable for hydrological modeling, especially in ungauged regions.

We sincerely thank the reviewer for this insightful comment, and we will provide a more detailed explanation of the novelty of our research in the Introduction and Discussion sections.

**Comments 2:** The structure of the paper needs to be reorganized. The Results section contains many texts that should be moved to Methods and Discussion. For example, Line 302-306 for how the optimal number of clusters is chosen should be moved to the Methods explaining FCM; Line 407-498 for the flow duration curve and hydrologic signature looks like a great point that should be moved to the Discussion section. Overall, the current organization, having discussions inside Results, makes the manuscript long and disruptive to read. The manuscript should be organized more neatly, where the Results should focus on presenting numbers, while moving and consolidating interpretations and implications in the Discussion.

**Response 2:** Thank you for your constructive suggestion. We greatly appreciate your feedback regarding the organization of our manuscript. We understand the importance of presenting the content in a clear and concise manner, and we agree that restructuring the paper would enhance its readability.

 In response to your comment, we will reorganize the manuscript by focusing on adjusting the structure of the Results and Discussion sections to improve its flow. We believe this reorganization will make the manuscript more streamlined and easier to follow. Thank you again for your helpful suggestion.

**Comments 3:** The validation of classification is only performed for 10 watersheds in all entire China, which I think is insufficient. Based on the Figure 6, there are many same-class watersheds that are fairly distant from each other. However, the current selection of watersheds, though the similarity of FDC in each class is shown, might be

insufficient to support the conclusion, as these watersheds in same classes are too spatially close to each other. Therefore, I am wondering how the similarity of FDC would be if watersheds that are more spatially distant are chosen for evaluation.

**Response 3:** Thank you for your valuable suggestion. We appreciate your concern regarding the validation of our classification, specifically the limited number of 10 watersheds across China. While these catchments are representative, their relatively close proximity may not fully capture the spatial variability within the same class.

In response, we will expand the validation by including more spatially diverse watersheds within the same climatic and landscape categories. This will allow us to better assess how the similarity of flow duration curves (FDCs) holds for geographically distant catchments and further validate the robustness of our classification framework.

Our manuscript already discusses the spatial heterogeneity observed across different climate regions, as seen in the significant differences in seasonal flow regimes and FDC curves (Fig. 8). While catchments within the same region often exhibit similar hydrological behaviors, there is substantial variability within the same class. We will highlight how spatially distant watersheds, located within the same climatic homogeneity region, contribute to validating the effectiveness of our classification method.

However, it is important to note that the availability of continuous hydrological data, particularly in remote areas, is a limitation that restricts the ability to include a larger number of spatially diverse catchments in the validation. Despite this constraint, we believe the validation performed with the available data provides valuable insights into the classification's performance.

Thank you again for your insightful suggestion, which will help strengthen the validation of our study.

**Comments 4:** The application of the study is not thoroughly discussed. The section 4.1 focuses more on the advantages of the probabilistic approach of FCM over hard-boundary classification ones, and the potential of improving regional hydrological modeling. However, the potential of 1) transferring model parameters from calibrated to ungauged watersheds and 2) estimating floods under various design storms based on the similarity of flow duration curve could be discussed, and can improve the novelty and value of the research.

**Response 4:** We sincerely appreciate this constructive suggestion. We agree that explicitly linking our classification framework to practical hydrological applications would significantly enhance the study's impact, especially for ungauged catchment management.

In response, we will expand Section 4.1 to include a more detailed discussion on the following applications:

Transferring Model Parameters from Calibrated to Ungauged Watersheds: Hydrological modeling in ungauged basins is a major challenge, particularly due to the lack of observational data for calibration. Transferring model parameters from calibrated watersheds to ungauged catchments is crucial for improving hydrological model performance in such regions. Our classification system, which groups watersheds with similar climatic and hydrological behaviors, offers a promising solution for this challenge. Specifically, we will illustrate how the Xinanjiang Model, a widely used hydrological model in China, can benefit from this approach.

We will discuss how our classification system allows for the transfer of model parameters from well-calibrated basins to ungauged catchments by identifying hydrologically similar watersheds. Additionally, we will emphasize how this method improves the transferability of model parameters across larger regions. This enhances the applicability of hydrological models in areas where observational data are sparse, ultimately facilitating more accurate simulations and predictions.

This added discussion will highlight how our approach can significantly improve regional hydrological modeling, particularly in regions where traditional calibration is difficult due to data scarcity. We believe these additions will demonstrate the practical applications and the novelty of our classification approach, making it more relevant for ungauged catchment management and improving flood prediction and water resource planning.

Thank you again for your valuable suggestion, which will help strengthen the practical contribution of our research.

**Comments 5:** I think the references are not in the required style of HESS (https://www.hydrology-and-earth-system-sciences.net/submission.html#references)

**Response 5:** Thank you for pointing this out. We apologize for the formatting issues with the references. We will carefully review the reference list and ensure that all citations are formatted according to the HESS style guidelines, as provided in the submission instructions. This will be corrected in the revised manuscript to meet the journal's requirements.

Thank you again for your attention to detail.

**Specific comments**

**Comments 1:** Line 69: "an indisputable fact" looks like a strange statement. Do you mean machine learning is now widely used for regionalization studies?

**Response 1:** Thank you for pointing out this concern. We agree that the phrase "an indisputable fact" may sound overly definitive. We meant to convey that machine learning is now widely applied and increasingly recognized in regionalization studies. To clarify this, we will revise the sentence to:

"*With the advancement of computer technology in the 21st century, the widespread use of machine learning in regionalization studies has become increasingly common and well-recognized (Yang et al., 2020a).*"We believe this revision more accurately conveys the intended meaning and avoids any ambiguity.

Thank you again for your constructive feedback.

**Comments 2:** Line 102-103: need to add references.

**Response 2:** Thank you for raising this important point regarding the need for references. We agree that supporting the statement with relevant literature will strengthen the argument. In response, we will add appropriate references to support the claim that climate patterns significantly influence the hydrological response of catchments, including their effects on soil moisture availability and the co-evolution of landscape and vegetation.

Thank you for your insightful feedback, which has helped us improve our work.

**Comments 3:** Line 117-126: The organization of this paragraph needs improvement. The six indices should be stated before reasoning why they are selected. It would make

the flow more logical, rather than making readers wonder what indices are chosen (line 119, three indices but not stating what they are).

**Response 3:** Thank you for your valuable suggestion. We appreciate your feedback regarding the organization of the paragraph. We agree that stating the six climate indices before explaining the reasoning for their selection would improve the flow and clarity of the discussion.

In response, we will restructure the paragraph to first list the six selected indices and then explain the rationale behind their selection. This will help readers understand the indices from the outset and follow the reasoning more logically. The revised paragraph will be as follows:

"*To capture the seasonal and spatial variability of temperature, we selected six indices for climate classification: average moisture index ($I_m$), seasonal moisture index ($I_{m,r}$), fraction that falls as snow (fs), annual average temperature ($T_m$), seasonal temperature ($T_{m,r}$), and the fraction of snowy days (Ds). The first three indices are expressed using a version of Thornthwaite's moisture index MI (Willmott and Feddema, 1992). These indices were chosen based on their ability to represent key climate factors, including moisture availability, snow distribution, and temperature, which are critical for understanding climate variability and hydrological responses. The indices were calculated for each $0.25°$ land cell using the CRU TS V4.04 dataset and meteorological station data. Although some of these indices have been previously used to map climate homogeneity regions, they have not been combined in this specific way.*"

This revision will make the paragraph more logical and easier to follow. Thank you again for your helpful suggestion.

**Comments 4:** Line 174: reference for FCM?

**Response 4:** Thank you for your question regarding the reference for fuzzy c-means clustering (FCM). We will include the appropriate reference for FCM to support the discussion. We will revise the sentence as follows: "*Fuzzy c-means clustering (FCM), based on fuzzy set theory, is one of the most widely used soft clustering algorithms (Bezdek, 1984). Unlike hard clustering algorithms, such as k-means and hierarchical clustering, the FCM cluster procedure uses a fuzzy parameter to create overlapping cluster boundaries.*"

We will update the manuscript to include this reference. Thank you again for your helpful suggestion.

**Comments 5:** Line 178: "may be the most" to "is a".

**Response 5:** Thank you for your suggestion. We agree that changing "may be the most" to "is a" will make the statement more definitive and clear. We will make this revision to improve the strength of the claim.

**Comments 6:** Line 207: reference for Penman-Monteith equation.

**Response 6:** Thank you for your comment regarding the reference for the Penman-Monteith equation. We will include the appropriate citation to support the use of this formula. We will revise the sentence to include the reference as follows:

"*The EP is estimated using a variant of the Penman-Monteith formula (Moratiel, 2020). The climate data were interpolated at a 0.25° × 0.25° spatial resolution, and missing data were filled using the weighted nearest-neighbor approach.*"

We will update the manuscript to include these references. Thank you again for your valuable suggestion.

**Comments 7:** Line 214: What are the average size and the range of catchment sizes?

**Response 7:** Thank you for your comment regarding the size of the catchments. Based on the HydroBASINS dataset, the average catchment size is 761.04 km², with the sizes ranging from 15 km² to 14,612.8 km². This information will be added to the manuscript to provide a clearer understanding of the spatial extent of the catchments analyzed.

**Comments 8:** Figure 2: Though the interpretations of hexagons values are provided, I still don't quite get the physical meanings of these plots and wonder how they should be interpreted spatially on maps (if a basemap can be added, it would be helpful). In the figure's caption, briefly explain the legends and how readers should interpret the figure. Also, line 276-285 should be moved to Methods

**Response 8:** Thank you for your valuable feedback. We appreciate your suggestion to enhance the interpretation of Figure 2. We will make the following improvements in response to your comments:

We will revise the figure caption to provide a clearer explanation of the physical meaning of the hexagonal values shown in Figure 2. Each hexagon represents a unit of

the self-organizing map (SOM) grid, with its color corresponding to the weight vector values of the climate indices. The color scale ranges from blue (low values) to red (high values), and we will explain that these color values represent the relative intensity of the climate indices across the SOM grid. We will also briefly mention how the component planes show spatial patterns of different climate variables, and how these patterns relate to the underlying catchment characteristics.

To improve the spatial interpretation, we will add labels for climate regions in Figure 2. This will help readers better understand the geographic distribution of the climate data and the relationship between climate regions and catchment characteristics. The figure caption will be updated to include a more detailed explanation of the legends and how to interpret the color-coded values for each hexagon. Specifically, we will explain how the color scale corresponds to the values of the climate indices and their spatial distribution across the SOM grid.

We agree that the explanation of the component planes and d-matrix in lines 276-285 would be more appropriate in the Methods section. We will move this content to ensure the Results section focuses on the analysis, while the Methods section provides the necessary background on the SOM and FCM algorithms and how they were applied.

Thank you again for your constructive suggestions, which will enhance the presentation and interpretation of the results.

**Comments 9:** Line 304: What is the AP algorithm? I don't think this was mentioned in Methods, and should add the reference

**Response 9:** Thank you for your valuable comment. We apologize for the confusion regarding the AP algorithm. The AP algorithm refers to the Affinity Propagation algorithm, a clustering algorithm that does not require the specification of the number of clusters in advance. It identifies cluster centers and assigns data points to the closest exemplar based on similarity.

We will add a brief explanation of the AP algorithm in the Methods section and include a citation to the relevant reference for this algorithm. The revised text in the Methods section will read as follows:

*"The Affinity Propagation (AP) algorithm (Frey & Dueck, 2007) was used to determine the maximum number of clusters. This algorithm identifies exemplar points in the*

*dataset and assigns other points based on similarity, without requiring the user to specify the number of clusters in advance."*

Additionally, we will update Section 3.1.3 to clarify this point:

"Before clustering the output neurons from the SOM competitive layer using the FCM algorithm, two validation metrics were calculated (i.e., DBI and SC) to determine the optimal number of clusters. An experiment was conducted to test the number of clusters from two to the maximum number determined by the Affinity Propagation (AP) algorithm (Shang, 2016)."

Thank you again for pointing this out. We will make these changes in the manuscript to provide clarity and ensure proper referencing.

**Comments 10:** Line 369: Two questions here: 1) For soil & veg characteristics, the second PC has an eigenvalue of 0.91. Why do you choose this PC below one? 2) Improve the writing: Instead of using semicolons, clearly state which class (topographic, soil & veg, and tolological) you are discussing first (For topographic, XXX. For soil & veg, XXX.), then state how each PC is correlated to the input indices.

**Response 10:** Thank you for your valuable feedback. We appreciate your questions and suggestions for improving the clarity of the manuscript. Here's how we will address your concerns:

According to the standard rule for PCA, we typically retain components with eigenvalues greater than 1. However, in this case, we kept the second principal component due to its significant contribution to the variance explained in the data (it helps explain a portion of the soil and vegetation characteristics, especially those that were not fully captured by the first component). While the eigenvalue is slightly below 1, the cumulative proportion of variance explained by the two components combined is greater than 70%, justifying the inclusion of both components. This will be clarified in the revised manuscript.

To improve clarity, we will revise the text to explicitly identify which feature class is being discussed before presenting the principal components' correlations. Instead of using semicolons, we will structure the sentences to state which class (topographic, soil & vegetation, and topological) is being discussed first, followed by the correlation of each principal component with the relevant indices.

Thank you again for your insightful suggestions, which will help improve the clarity and robustness of our study.

**Comments 11:** Line 449: space between class and (Li et al).

**Response 11:** Thank you for pointing out the formatting issue. We will add a space between "class" and the citation "(Li et al., 2018)" to ensure proper formatting.

**Comments 12:** Figure 7: some recommendations: 1) List each site/catchment's classification in the line charts, and 2) maybe consider another color scheme to present the variation range. The grey colors are hard to differentiate. Also, briefly describe the ranges in the legend within the caption for people to understand.

**Response 12:** Thank you for your valuable feedback and suggestions regarding Figure 7. Here's how we will address your comments:

We will add labels to the line charts to clearly indicate the classification of each catchment. This will help readers easily identify the site corresponding to each line, improving the clarity of the figure. We agree that the grey colors are difficult to differentiate. We will consider using a more distinct and visually accessible color scheme to present the variation range. This will make it easier for readers to distinguish between the different percentiles (e.g., 10%, 50%, 90%). We will modify the figure caption to briefly describe the ranges represented by the colors in the legend. This will help readers better understand the variation in runoff across the different catchments and regions.

Thank you again for your helpful suggestions, which will enhance the figure's clarity and overall presentation.

**Comments 13:** Line 495: correct the citation

**Response 13:** Thank you for pointing out the issue. We apologize for the oversight. We will revise the sentence as follows:

*"Additionally, catchments with large spatial distances are capable of exhibiting similar hydrological characteristics (e.g., Maduwang and Daiying)."*

We will ensure that the correct context is clearly presented in the manuscript.

Thank you again for your attention to detail.

**Comments 14:** Figure 8: What will FDCs look like if using discharge in mm/day (normalized by drainage area)? I am wondering this because these watersheds vary significantly in drainage size, and normalizing discharge by size may allow for expanding the validation to more gauged watersheds.

**Response 14:** Thank you for your insightful suggestion regarding the normalization of discharge by drainage area. We agree that normalizing discharge by the catchment area could be a valuable approach. By normalizing the discharge in units of mm/day (precipitation depth per day), the FDCs would become more comparable across watersheds of different sizes, allowing for a more consistent validation across a broader range of gauged catchments.

We will update the manuscript to include a discussion of this potential modification. Specifically, we will mention that normalizing discharge by drainage area could allow for extending the validation of our classification method to more gauged watersheds, especially those with different catchment sizes.

Thank you again for your valuable suggestion, which will help enhance the robustness of our analysis.

**Comments 15:** Line 517: The statement, "leading to errors", should be more evidence-based. What specific errors could inaccurate classification result in? What consequences/risks will these errors cause? Provide references of previous studies showing so.

**Response 15:** Thank you for your valuable comment. We agree that the statement, "leading to errors," should be more evidence-based. Inaccurate classification due to the introduction of artificial boundaries can lead to several errors in hydrological modeling:

Artificial boundaries may cause misclassification of hydrological regions, leading to inaccurate parameterization of models. For instance, a catchment classified in the wrong climatic zone may have model parameters (e.g., runoff coefficients, baseflow, etc.) that do not properly reflect its hydrological processes, resulting in less accurate predictions of runoff and streamflow. Misclassification at the boundaries of climatic zones could lead to incorrect predictions of flood events, as the flow regimes and seasonal patterns may be poorly represented. This could significantly affect flood risk assessment and the design of flood protection measures.

We will revise the manuscript to incorporate these explanations, along with relevant references, to strengthen the evidence supporting our claim. Thank you again for your insightful comment, which will improve the clarity and depth of our manuscript.

**Comments 16:** Line 546: Is this a possible reason causing the challenge that some watersheds are hard to be classified with one dominant group?

**Response 16:** Human activities, such as urbanization and agriculture, can make it more challenging to classify some watersheds into one dominant group. These human influences can cause significant deviations in hydrological behavior, even among watersheds with similar climatic and topographical characteristics. As a result, catchments with similar natural features may exhibit different hydrological responses due to anthropogenic factors, complicating the classification process.

This variability can contribute to the challenge of grouping certain watersheds into a single, dominant cluster. We emphasize the importance of considering both natural and anthropogenic factors in the classification process, which requires a more comprehensive analysis of watershed characteristics. We will revise the manuscript to include this explanation, highlighting the importance of integrating human factors into the classification process in future studies.

Thank you again for your valuable suggestion, which will enhance the clarity and depth of our discussion.

---

## Author Comment (AC4)

**Major Comments**

**Comments 1:** The Results section should focus on reporting experimental outcomes and data interpretation only. Methodological explanations, such as the description of the Affinity Propagation (AP) algorithm (Line 304), should be moved to the Methods section for better structural consistency.

**Response 1:** Thank you very much for your constructive suggestion. We greatly appreciate your feedback regarding the organization of our manuscript. We fully understand the importance of presenting the content in a clear and concise manner. In response to your comment, we will revise the manuscript to ensure that the Results section focuses exclusively on reporting experimental outcomes and data interpretation. The methodological explanation of the Affinity Propagation (AP) algorithm (currently at Line 304) will be moved to the Methods section to achieve better structural consistency and logical coherence.

Furthermore, we will refine the overall structure of the Results and Discussion sections to enhance the manuscript's readability and flow. We believe that these planned revisions will make the paper more concise and easier to understand. Once again, we sincerely thank you for your valuable and helpful advice.

**Comments 2:** Please justify the representativeness of these catchments or consider expanding the validation dataset to reinforce the robustness of your conclusions.

**Response 2:** Thank you very much for this insightful comment. We appreciate the reviewer's concern regarding the representativeness of the selected small catchments used for validation. We fully agree that clarifying the representativeness of these catchments or expanding the validation dataset can further strengthen the robustness of our conclusions.

In response to this comment, we will revise the Validation Results section (Section 3.3) to explicitly justify the selection of the ten representative catchments. Specifically, we will provide additional details on their spatial distribution, climatic diversity, and geomorphological variability to demonstrate that they adequately represent the major climate regions and catchment types identified through the SOM – FCM classification framework.

We also fully agree that expanding the validation dataset would further enhance the reliability of the results. However, due to the practical challenges in obtaining longterm, high-quality, and natural (unregulated) runoff data—one of the core motivations behind focusing on ungauged regions in this study—a large-scale expansion of the validation dataset is currently difficult. Therefore, we will include a clear statement in the Discussion or Conclusion section identifying dataset expansion as an important future research direction. Specifically, we will point out that future work could extend the validation to additional climate regions to further examine the generalizability and robustness of the proposed classification framework.

We sincerely thank the reviewer for this valuable suggestion, which will help to strengthen the scientific rigor and credibility of our study.

**Comments 3:** Please include a paragraph discussing the limitations of the proposed method and outline possible future research directions to improve applicability and generalization.

**Response 3:** Thank you very much for this constructive suggestion. We fully agree with the reviewer that discussing the limitations of the proposed method and outlining potential future research directions will improve the completeness, transparency, and scientific rigor of our study.

In response, we will add a dedicated paragraph in the Discussion and Conclusion sections to explicitly address these limitations and outline future work. Specifically, we will acknowledge the following points:

**(1) Limitations related to static catchment attributes and data availability:**

Our classification framework is based on long-term average hydroclimatological and geomorphological attributes, which inherently represent a static perspective of catchment characteristics. It does not explicitly account for dynamic processes such as land use and land cover change, anthropogenic impacts (e.g., increasing water withdrawals), or long-term climate shifts. Moreover, the resolution and accuracy of the classification depend on the quality and spatial resolution of the underlying national datasets. In regions with sparse gauge networks, data limitations may introduce additional uncertainty.

**(2) Limitations in validation dataset coverage:**

Although the selected validation catchments are representative, their spatial coverage remains limited due to the availability of long-term, high-quality, and natural runoff

records. This restricts the full assessment of classification performance across all climate regions.

Furthermore, we will outline future research directions, including: (1) integrating more dynamic hydrological variables (e.g., evapotranspiration, snowmelt indices) to enhance model interpretability and physical realism; (2) testing the transferability of the classification across additional climate zones and transboundary basins to evaluate generalization; and (3) coupling the classification framework with process-based hydrological models to improve simulation performance and applicability in ungauged catchments. We believe that including this discussion will provide a more balanced and comprehensive assessment of the proposed methodology, while clearly identifying pathways for its future improvement and broader applicability.

**Comments 4:** I think you need check the English grammar and sentence carefully. Please revise the English description of your manuscript.

**Response 4:** Thank you very much for your careful reading and valuable comment. We appreciate the reviewer's attention to the language quality of our manuscript. We fully agree that clear and precise English expression is essential for ensuring the readability and professionalism of the paper.

In response to this comment, we will carefully review the entire manuscript to correct grammatical errors, improve sentence structure, and enhance the overall clarity and fluency of the English text. We sincerely thank the reviewer for pointing this out, which will help us further improve the overall presentation and readability of our work.

**Comments 5:** Please carefully review and standardize all references to conform with the HESS reference style guide.

**Response 5:** Thank you very much for this helpful comment. We appreciate the reviewer's careful attention to the consistency and accuracy of our references. We fully agree that ensuring all references strictly adhere to the Hydrology and Earth System Sciences (HESS) reference style is essential for maintaining the quality and professionalism of the manuscript.

In response to this comment, we will carefully review all references in the revised manuscript to ensure that they conform precisely to the HESS reference style guide.

**Minor Comments**

Comments 1: Line 276: When describing the SOM neuron grid  $(19 \times 22)$ , briefly justify why this specific grid size was selected (e.g., based on data dimensionality, heuristic optimization, or quantization error minimization).

**Response 1:** Thank you very much for this insightful comment. We appreciate the reviewer's attention to the justification of the SOM neuron grid configuration.

In response, we will emphasize the rationale for selecting the grid size in the Methods section. Specifically, we will explain that the  $19 \times 22$  SOM structure was determined through a combination of heuristic optimization and internal network performance evaluation. The grid dimensions were selected by testing multiple configurations and assessing their quantization error (QE) and topographic error (TE) values. The  $19 \times 22$  grid achieved the optimal balance—minimizing QE and TE while maintaining a sufficient number of neurons to capture the spatial variability and complexity of the input climate data.

This clarification will make the methodological rationale for the SOM configuration more transparent and reproducible.

**Comments 2:** Figure 2: The description of the SOM component planes is clear but would benefit from a short explanatory note in the figure caption clarifying the color scale meaning (e.g., "red indicates high values, blue indicates low values").

**Response 2:** Thank you very much for this helpful comment. We appreciate the reviewer's suggestion to improve the clarity of Figure 2. In response, we will revise the figure caption to include a short explanatory note clarifying the color scale meaning in the SOM component planes. This addition will help readers more intuitively interpret the component planes and better understand the spatial patterns represented by different neurons in the SOM output.

**Comments 3:** Ensure consistent use of "" or "Figure" throughout the manuscript according to the journal's style guide.

**Response 3:** Thank you very much for your careful observation. In response, we will thoroughly review the entire manuscript to ensure the consistent use of the term "Figure" (or its abbreviated form "Fig.") in accordance with the HESS journal style guide.

All figure citations in the text will be standardized to follow the required format, ensuring stylistic consistency and improving the overall presentation quality of the manuscript.

Comments 4: The transition between Section 3.1.3 (FCM clustering results) and Section 3.2 (Results of catchment classification) is abrupt. Consider adding a bridging sentence such as: "Based on the derived climate clusters, we further classified catchments with similar landscape attributes within each climate region."

**Response 4:** Thank you very much for this constructive comment. We fully agree that adding a bridging sentence will improve the logical flow and readability of the manuscript. In response, we will revise the transition between these two sections by adding a short linking sentence to clearly indicate the methodological connection between the climate-based clustering and the subsequent catchment classification. Specifically, we will add the following sentence at the beginning of Section 3.2 "Based on the derived climate clusters, we further classified catchments with similar landscape attributes within each climate region."

This addition will create a smoother transition and help readers better understand how the classification framework progresses from climate clustering to catchment-level differentiation.

**Comments 5:** Lines 460 – 471:The sentence "The flow regime in climate region II presented multiple peaks following multiple peaks in precipitation in June and July during the same period." is ambiguous. Please revise or clarify its intended meaning.

**Response 5:** Thank you for your valuable feedback on our manuscript. We agree that this section could be more clearly written to improve readability and ensure that the findings are easily understood. In response, we will revise the paragraph in Lines 460 – 471 to provide a clearer description of the seasonal flow regime and its relationship to precipitation. Specifically, we will rephrase the sentence to more precisely express that the runoff in climate region II exhibits multiple seasonal peaks corresponding to successive precipitation peaks in June and July. This clarification will make the relationship between rainfall patterns and flow responses more explicit and easier for readers to interpret.

Thank you again for your feedback, and we will make the necessary improvements based on your suggestions.

**Comments 6:** Line 494:The expression "Additionally, catchments with large spatial distances are capable of exhibiting similar hydrological characteristics (Maduwang and Daiying, year)" needs revision for clarity and proper citation formatting.

**Response 6:** Thank you for pointing out the issue. We apologize for the oversight. We will revise the sentence as follows: "Additionally, catchments with large spatial distances are capable of exhibiting similar hydrological characteristics (e.g., Maduwang and Daiying)."

We will ensure that the correct context is clearly presented in the manuscript. Thank you again for your attention to detail.

**Comments 7:** Please revised your figure and make it clearly, specially the size and format of figures.

**Response 7:** We sincerely thank the reviewer for highlighting the need to improve the quality of our figures. We fully agree that clear and well-designed figures are essential for effectively communicating our results. Accordingly, we will carefully check all figures and tables to ensure they meet the journal's formatting and quality standards.